# Do LLMs Understand Social Knowledge? Evaluating the Sociability of Large Language Models with the SOCKET Benchmark

**Minje Choi**[†*] **Jiaxin Pei**[†*] **Sagar Kumar** [‡] **Chang Shu**[♯] **David Jurgens**[†]

[†]University of Michigan, Ann Arbor, MI, USA
[‡]Northeastern University, Boston, MA, USA
[♯] University of Cambridge, Cambridge, UK
♣{minje, pedropei, jurgens}@umich.edu
[†]kumar.sag@northeastern.edu [‡]cs2175@cam.ac.uk

## Abstract

Large language models (LLMs) have been shown to perform well at a variety of syntactic, discourse, and reasoning tasks. While LLMs are increasingly deployed in many forms including conversational agents that interact with humans, we lack a grounded benchmark to measure how well LLMs understand *social* language. Here, we introduce a new theory-driven benchmark, SOCKET, that contains 58 NLP tasks testing social knowledge which we group into five categories: humor & sarcasm, offensiveness, sentiment & emotion, trustworthiness, and other social factors. In tests on the benchmark, we demonstrate that current models attain only moderate performance but reveal significant potential for task transfer among different types and categories of tasks, which were predicted from theory. Through zero-shot evaluations, we show that pretrained models already possess some innate but limited capabilities of social language understanding and training on one category of tasks can improve zero-shot testing on others. Our benchmark provides a systematic way to analyze model performance on an important dimension of language and points to clear room for improvement to build more socially-aware LLMs. The resources are released at https://github.com/minjechoi/SOCKET.

## 1 Introduction

Interpersonal communication is more than just what is said. Understanding communication requires reasoning not only about the content of a message but also the social implications drawn from that message (Halliday, 1995). As NLP systems, particularly Large Language Models (LLMs), are increasingly used in interpersonal settings, these models' abilities to understand social knowledge become critical. However, despite the recognized need for social knowledge (Hovy and Yang,

2021), the NLP field has limited abilities to test it. Here, we introduce SOCKET, a new benchmark for evaluating social knowledge.

Evaluating NLP systems has remained a key component for benchmarking the field's progress. Indeed, the rapid replacement of traditional models by LLM-based approaches was strongly motivated by substantial gains by LLMs on a variety of comprehensive Natural Language Understanding (NLU) benchmarks like SuperGLUE (Wang et al., 2019) and Natural Questions (Kwiatkowski et al., 2019). However, despite the fundamental social aspect of language, comprehensive benchmarks of social language remain absent. Instead, existing computational studies of social language have built individual datasets and models for specific types of information like empathy (Sharma et al., 2020), politeness (Danescu-Niculescu-Mizil et al., 2013), and humor (Van Hee et al., 2018). While beneficial, these semantic-level tasks omit broader social and narrative-level information (Li et al., 2021) and present only a narrow view of model performance.

We introduce SOCKET (**Soc**ial **K**nowledge **E**valuation **T**ests), a theory-grounded, systematic collection of 58 social language tasks.[1] SOCKET covers five categories of social information: sentiment & emotion, trustworthiness, humor & sarcasm, offensiveness, and social factors, each motivated by specific theories. To examine models' generalizability, SOCKET includes four task formats: classification, regression, pairwise comparison, and span identification. This construction aims at assessing not only NLP models' performances on individual tasks but their ability to perform multiple task types and to productively benefit from related tasks and task categories during learning.

Our study offers the following three contribu-

---

*equal contribution

[1]The choice of the term "social knowledge" in framing stems from its use for a broad category in psychology (e.g., Turiel, 1983; Adolphs, 2009) that matched the capabilities we are interested in.

tions to the research community. (1) We motivate a theoretically-grounded organization of social tasks (§2) and subsequently introduce a new easy-to-use benchmark, SOCKET, that systematically organizes 58 tasks (§3). (2) We benchmark multiple current LLM approaches to multitask NLU via standard supervised training and zero-shot LLMs (§4). Across all tests, our results show that baseline LLMs perform moderately, at best, but offer promising signs of being able to leverage task correlations. (3) We test the abilities of models to make use of cross-task transfer (§5) showing multi-task training on strongly correlated tasks can maintain or even improve performance in specific tasks, but doing so on weakly correlated tasks can hurt the overall performance of LLMs (§6). We release our framework code and prepackaged datasets at `https://github.com/minjechoi/SOCKET` and `https://huggingface.co/datasets/Blablablab/SOCKET`.

## 2 Social Information in Natural Language Processing

Language is inherently social, as meaning is constructed through social interactions (Wittgenstein, 1953). A substantial body of research in linguistic theory and communication studies have examined how social knowledge is communicated via language understanding. Theories of language grounded in interaction and communication systems such as Systemic Functional Linguistics (SFL) by Halliday et al. (1989) assert that the function and appropriacy of language in a given context is the key to our understanding of language and its use (Eggins, 2004; Allan, 2007; Halliday et al., 1989; Halliday, 2004). We use these insights to probe linguistic models for their ability to capture *social information*, which we define as information conveyed through text about broader metatextual function and contextual appropriacy of the utterances in conversation.

**NLP Studies on Social Information** Numerous studies have contributed to the development of datasets and models aimed toward identifying nuanced social information in language across diverse contexts. Computational linguists have modeled multiple forms of social information in language like sentiment (Buechel and Hahn, 2017), politeness (Fu et al., 2020), humor (Meaney et al., 2021), offensiveness (ElSherief et al., 2021), and intimacy (Pei and Jurgens, 2020), often achieving state-of-

the-art results close to human performance in their respective settings. Studies such as Park et al. (2021) have also leveraged explicitly-given norms to train models to be more accurate in context-specific situations.

However, these plausible results may be achievable solely by focusing on the statistical and syntactical instead of the social aspects of language. Whether to make advances in language understanding in research or to ensure reliability and safety in deployment, it is of vital importance to study whether models are truly capable of gaining a generalizable understanding of social factors before employing them for tasks that require such knowledge (Hovy and Yang, 2021). The necessity for such understanding is exemplified by studies showing that, when measuring the same concept, the performance of a model can vary greatly when tested on a different dataset due to factors such as changes in dialect, speaker demographics, and dataset domain (Miller et al., 2020; Blodgett et al., 2016; Wang et al., 2022a).

Despite this importance, efforts towards aggregating and synthesizing various datasets into themes have been less practiced. One notable exception is the work of Kang and Hovy (2021), where the authors combine existing datasets on different linguistic styles to introduce a benchmark that enables them to study cross-style language understanding. Similarly, we present a benchmark curated from over fifty different tasks on different aspects of social information, which we group into five distinctive categories.

**Examining the social knowledge of LLMs** LLMs are ubiquitous in NLP and their success is attributed to the ability to capture language characteristics from the immense amount of text seen in pre-training and to effectively apply this information on downstream tasks, achieving state-of-the-art performances in many language understanding tasks (Chung et al., 2022a). LLMs have demonstrated less success when solving tasks directly related to social knowledge. For tasks that require social information such as detecting sarcasm (Farha et al., 2022) or patronizing language (Perez-Almendros et al., 2022), recent models exhibit only moderate performance. One major challenge is that compared to humans, LLMs have less capability to make predictions outside of the provided input and must perform reasoning only based on their innate social information (Sap et al.,

2019b; Zhou et al., 2020). Yet, it is this very social knowledge that is crucial for human interactions and conversations and is a milestone that should be reached for LLMs to engage in meaningful communications with humans (Mahowald et al., 2023).

More recently, general-purpose LLMs trained with instruction-based prompts have been known to achieve strong performances, putting them to use in several practical domains such as summarization, question answering, and classification (Sanh et al., 2022). A newly emerging trend is to use curated prompts to identify the psychological capabilities of instruction-guided LLMs. Ruis et al. (2022) and Hu et al. (2022a) examine pragmatic understanding capabilities using prompts. Coupled with additional steps such as chain-of-thought (CoT) reasoning, this prompt-based approach has large potential for understanding whether LLMs can provide reasoning capabilities like humans.

**The Inter-relatedness of Social Information** Social language understanding requires accurately perceiving different dimensions and facets of communication that relate to one another. Interpersonal communication makes frequent use of humor (Schnurr, 2010), mitigation, also known as hedging, (Schneider, 2010), and swearing as a norm violation (Stapleton, 2003) in defining the contours of the social context for the speakers. Often, the pragmatics of these different dimensions of social language use are intertwined: communication with one dimension influences the interpretation of another, e.g., politeness and offensive speech (Culpeper, 2021), humor and politeness (Attardo, 2008), humor and offensiveness (Alberts, 1992), and mitigation and empathy (LI Hai-hui, 2019). Understanding one of these dimensions requires models to have the ability to recognize the related dimensions. While past computational work has largely focused on single dimensions, SOCKET fills a key gap by testing whether models can accurately recognize multiple, interrelated social dimensions—and whether models can benefit in their understanding from cross-task transfer.

## 3 The SOCKET Benchmark

Here, we describe the steps taken to curate SOCKET as robust benchmark for identifying social information embedded in language in interpersonal communication contexts.

### 3.1 Task Selection Process

The task curation process began with a systematic review of literature on social from linguistics, communications, and psychology to identify likely categories of social knowledge. Then, possible datasets and tasks were identified through a systematic review of datasets published at ACL, EMNLP, NAACL, EACL, LREC, and SemEval since 2015. In this first pass, we selected more than 100 datasets and tasks to detect different types of social information in language (cf. Table 11 in Appendix B.9 for all candidate datasets and tasks). Tasks were selected based on membership in five categories of social language (described next) that are motivated as core aspects of social language understanding.

For each category, we include tasks of several distinct objectives: binary and multi-class classification, regression, pairwise similarity detection, and span identification.[2] Where possible, we aim for diversity within categories and ensure one task for each objective. Candidate tasks were removed if it was found that training a `bert-base-uncased` model on the task achieved test performance over 0.95, which would provide little insight into progress at recognizing social information .

While this process identified many candidate tasks in multiple categories, the benchmark still defines only partial progress in social knowledge capabilities. Some abilities recognized by social sciences such as deceit have only one or two tasks proposed (Ott et al., 2011), providing limited data to measure progress. However, recognizing these as limitations (discussed in more detail in §8), SOCKET provides a diverse set of tasks and capabilities, described next, for the field to begin to measure progress.

### 3.2 Task categories

Inspired by theories in interpersonal communication and interpersonal pragmatics, we provide a thematic organization of the tasks in SOCKET into five related categories of social knowledge: Humor & Sarcasm, Offensiveness, Sentiment & Emotion, Social Factors, and Trustworthiness.

**Humor & Sarcasm** The practice of humor in conversations and interactions plays a key role

---

[2]Other task types were initially considered (e.g., generation, paraphrasing) but such tasks were not feasible for all models and often were less standardized in their evaluation, complicating cross-task comparison if included.

| category | dataset | task name | size | type | labels | category | dataset | task name | size | type | labels |
|---|---|---|---|---|---|---|---|---|---|---|---|
| Humor & Sarcasm | hahackathon (Meaney et al., 2021) | humor_rating | 6,179 | REG | RMSE | Sentiment & Emotion | crowdflower (CrowdFlower, 2016) | crowdflower | 40,000 | CLS | 13 (F1) |
| Humor & Sarcasm | humor-pairs (Hossain et al., 2020) | humor-pairs | 15,095 | PAIR | 2 (F1) | Sentiment & Emotion | dailydialog (Li et al., 2017) | dailydialog | 102,979 | CLS | 7 (F1) |
| Humor & Sarcasm | sarc (Khodak et al., 2018) | sarc | 321,748 | CLS | 2 (F1) | Sentiment & Emotion | emobank (Buechel and Hahn, 2017) | arousal | 10,062 | REG | MAE |
| Humor & Sarcasm | tweet_irony (Van Hee et al., 2018) | tweet_irony | 4,601 | CLS | 2 (F1) | Sentiment & Emotion | emobank (Buechel and Hahn, 2017) | dominance | 10,062 | REG | MAE |
| Humor & Sarcasm | hahackathon (Meaney et al., 2021) | is_humor | 10,000 | CLS | 2 (F1) | Sentiment & Emotion | emobank (Buechel and Hahn, 2017) | valence | 10,062 | REG | MAE |
| Offensiveness | contextual-abuse (Vidgen et al., 2021) | IdentityDirectedAbuse | 13,450 | CLS | 2 (F1) | Sentiment & Emotion | emotion-span (Ghazi et al., 2015) | emotion-span | 820 | SPAN | 3 (F1) |
| Offensiveness | contextual-abuse (Vidgen et al., 2021) | PersonDirectedAbuse | 13,450 | CLS | 2 (F1) | Sentiment & Emotion | empathy (Buechel et al., 2018) | distress | 1,859 | REG | Corr. |
| Offensiveness | hahackathon (Meaney et al., 2021) | offense_rating | 10,000 | REG | RMSE | Sentiment & Emotion | empathy (Buechel et al., 2018) | distress_bin | 1,859 | CLS | 2 (F1) |
| Offensiveness | hasbiasedimplication (Sap et al., 2020) | hasbiasedimplication | 44,781 | CLS | 2 (F1) | Sentiment & Emotion | same-side-pairs (Körner et al., 2021) | same-side-pairs | 175 | PAIR | 2 (F1) |
| Offensiveness | hateoffensive (Davidson et al., 2017) | hateoffensive | 24,783 | CLS | 3 (F1-M) | Sentiment & Emotion | sentitreebank (Socher et al., 2013) | sentitreebank | 119,794 | CLS | 2 (Acc.) |
| Offensiveness | implicit-hate (ElSherief et al., 2021) | explicit_hate | 21,476 | CLS | 2 (F1) | Sentiment & Emotion | tweet_emoji (Barbieri et al., 2018) | tweet_emoji | 100,000 | CLS | 20 (F1-M) |
| Offensiveness | implicit-hate (ElSherief et al., 2021) | implicit_hate | 21,476 | CLS | 2 (F1) | Sentiment & Emotion | tweet_emotion (Mohammad et al., 2018) | tweet_emotion | 5,052 | CLS | 4 (F1-M) |
| Offensiveness | implicit-hate (ElSherief et al., 2021) | incitement_hate | 21,476 | CLS | 2 (F1) | Sentiment & Emotion | tweet_sentiment (Rosenthal et al., 2017) | tweet_sentiment | 59,899 | CLS | 3 (AvgRec) |
| Offensiveness | implicit-hate (ElSherief et al., 2021) | inferiority_hate | 21,476 | CLS | 2 (F1) | Social Factors | complaints (Preoţiuc-Pietro et al., 2019) | complaints | 3,449 | CLS | 2 (F1) |
| Offensiveness | implicit-hate (ElSherief et al., 2021) | stereotypical_hate | 21,476 | CLS | 2 (F1) | Social Factors | empathy (Buechel et al., 2018) | empathy | 1,859 | REG | Corr. |
| Offensiveness | implicit-hate (ElSherief et al., 2021) | threatening_hate | 21,476 | CLS | 2 (F1) | Social Factors | empathy (Buechel et al., 2018) | empathy_bin | 1,859 | CLS | 2 (F1) |
| Offensiveness | implicit-hate (ElSherief et al., 2021) | white_grievance_hate | 21,476 | CLS | 2 (F1) | Social Factors | hayati_politeness (Hayati et al., 2021) | hayati_politeness | 320 | CLS | 2 (F1) |
| Offensiveness | intentyn (Sap et al., 2020) | intentyn | 44,781 | CLS | 2 (F1) | Social Factors | questionintimacy (Pei and Jurgens, 2020) | questionintimacy | 2,247 | REG | 6 (Corr.) |
| Offensiveness | jigsaw (Jigsaw, 2017) | severe_toxic | 200,703 | CLS | 2 (F1) | Social Factors | stanfordpoliteness (Fu et al., 2020) | stanfordpoliteness | 10,956 | CLS | 2 (MAE) |
| Offensiveness | jigsaw (Jigsaw, 2017) | identity_hate | 200,703 | CLS | 2 (F1) | Trustworthiness | bragging (Jin et al., 2022) | brag_achievement | 6,643 | CLS | 2 (F1) |
| Offensiveness | jigsaw (Jigsaw, 2017) | threat | 200,703 | CLS | 2 (F1) | Trustworthiness | bragging (Jin et al., 2022) | brag_action | 6,643 | CLS | 2 (F1-M) |
| Offensiveness | jigsaw (Jigsaw, 2017) | obscene | 200,703 | CLS | 2 (F1) | Trustworthiness | bragging (Jin et al., 2022) | brag_possession | 6,643 | CLS | 2 (F1-M) |
| Offensiveness | jigsaw (Jigsaw, 2017) | insult | 200,703 | CLS | 2 (F1) | Trustworthiness | bragging (Jin et al., 2022) | brag_trait | 6,643 | CLS | 2 (F1-M) |
| Offensiveness | jigsaw (Jigsaw, 2017) | toxic | 200,703 | CLS | 2 (F1) | Trustworthiness | hypo-l (Zhang and Wan, 2022) | hypo-l | 3,226 | CLS | 2 (F1) |
| Offensiveness | offensiveyn (Sap et al., 2020) | offensiveyn | 44,781 | CLS | 2 (F1) | Trustworthiness | neutralizing-bias-pairs (Pryzant et al., 2020) | neutralizing-bias-pairs | 93,790 | PAIR | 2 (Acc.) |
| Offensiveness | sexyn (Sap et al., 2020) | sexyn | 44,781 | CLS | 2 (F1) | Trustworthiness | propaganda-span (Martino et al., 2020) | propaganda-span | 357 | SPAN | 3 (F1-m) |
| Offensiveness | talkdown-pairs (Wang and Potts, 2019) | talkdown-pairs | 6,510 | PAIR | 2 (F1) | Trustworthiness | rumor (Ma et al., 2017) | rumor_bool | 1,417 | CLS | 2 (F1) |
| Offensiveness | toxic-span (Pavlopoulos et al., 2021) | toxic-span | 10,621 | SPAN | 3 (F1) | Trustworthiness | two-to-lie (Peskov et al., 2020) | receiver_truth | 11,728 | CLS | 2 (F1-M) |
| Offensiveness | tweet_offensive (Zampieri et al., 2019b) | tweet_offensive | 14,100 | CLS | 2 (F1) | Trustworthiness | two-to-lie (Peskov et al., 2020) | sender_truth | 11,728 | CLS | 2 (F1-M) |

Table 1: A list of the datasets covered in the SOCKET benchmark. A total of 58 tasks in 5 categories of social information. Included are each task's sample size, task type and evaluation metric used in the original paper. SOCKET covers four types of tasks: classification (CLS), regression (REG), pair-wise comparison (PAIR), and span identification (SPAN). F1, F1-M and F1-m indicate binary F1, macro F1 and micro F1 scores.

in maintaining and forming positive social relations (Holmes, 2006; Brown et al., 1987; Ziv, 2010). We differ Humor & Sarcasm from Trustworthiness as a social information category because while both categories consider non-cooperative behaviors (Grice, 1975), humor is considered to be prosocial (Attardo, 2008). In instances where the humor is not considered to be prosocial and is instead of a derogatory nature, we consider it to be in the Offensiveness category. By nature, humor is a subjective concept that can differ depending on both demographic and contextual factors (Ruch, 2010), making humor detection a difficult task for LLMs. SOCKET includes a number of tasks on humor that can occur in various contexts such as in social media (Meaney et al., 2021), short jokes (Meaney et al., 2021), and news headlines (Hossain et al., 2020). We also include tasks that require detecting relevant concepts of humor such as sarcasm (Khodak et al., 2018) and irony (Van Hee et al., 2018).

**Offensiveness** Detecting offensiveness using computational methods has gained large attraction in recent years due to the ubiquity of online communication and the necessity to implement automated content moderation to combat abusive behaviors (Spertus, 1997). However, most existing studies only focus on limited types of offensive languages (Jurgens et al., 2019). In this study, we consider offensiveness to be any explicit or implicit language directed towards individuals, entities, or groups (Waseem et al., 2017), and the tasks chosen are representative of this understanding. SOCKET includes a list of offensiveness detection tasks covering different levels of harmful content and abusive language including both explicit and implicit

hate (ElSherief et al., 2021), abuse (Vidgen et al., 2021), and humor-related offensiveness (Meaney et al., 2021). We also include forms of bias directed towards people and groups, as social bias enforces harmful stereotypes (Sap et al., 2020).

**Sentiment & Emotion** Emotion is a core element of interpersonal communication that can be communicated through human language in several aspects (Majid, 2012; Barrett et al., 2007). Social information is crucial in the ability to not only communicate, but also feel emotion. Theories of discretized emotion (Ekman, 1992) have been supported by empirical findings that humans use discrete labels learned through language to direct their emotional responses to stimuli (Lindquist and Barrett, 2008). Moreover, emotional responses have been shown to direct communication with peers (Lee et al., 2020), and expressing certain emotional responses—such as anger—have been shown to have social ramifications (Keltner et al., 1993). Interpreting emotions from text using computational tools has been a popular research topic across numerous areas in social sciences, enabling new discoveries at unprecedented scale (Jackson et al., 2022). In SOCKET, we include a wide range of tasks from various domains such as daily dialogue (Li et al., 2017), written responses to news stories (Buechel and Hahn, 2017), and tweets using textual syntax (Mohammad et al., 2018), and also emojis (Barbieri et al., 2018).

**Trustworthiness** People can detect cues in language that determine the trustworthiness of a message (Newman et al., 2003), leading to studies that aim to quantify the level of trust in text using computational methods (Choi et al., 2020). In particu-

lar, this direction has gained attention from NLP communities following increased needs to combat and mitigate potential harms coming from the generation and dissemination of false information in online spaces (Wu et al., 2019). In SOCKET we include tasks that require identifying perceived trust from several dimensions: impartiality (Pryzant et al., 2020), deception (Ott et al., 2011), propaganda (Martino et al., 2020), rumor (Ma et al., 2017) and bragging, as it is considered to be "unplain speaking" (Haiman, 1998; Jin et al., 2022).

**Other Social Factors** Finally, we include tasks of a more discursive and rhetorical type, that are understood to be more reliant on the contextual elements of social distance, power, and solidarity. In SOCKET, the tasks included are empathy (Buechel et al., 2018), politeness (Hayati et al., 2021; Fu et al., 2020), intimacy (Pei and Jurgens, 2020) and complaints (Preoţiuc-Pietro et al., 2019). Politeness, like humor, is understood to be a non-cooperative prosocial behavior but unlike humor, is concerned with the act of "saving face" (Brown and Levinson, 1987). Empathy, shown to be closely related to politeness (Fukushima and Haugh, 2014), is heavily reliant on social positions in the context of the conversation (Macagno et al., 2022). Intimacy, however, has been shown to be more dependent on notions of time and space between people in dialogue (Márquez Reiter and Frohlich, 2020).

### 3.3 Dataset Summary

The final SOCKET benchmark contains 58 tasks from 35 datasets, grouped into the five categories shown in Figure 1. We denote multiple tasks from the same dataset by adding the task name as a suffix following the dataset name and # symbol.

The collection of tasks chosen for SOCKET makes it a comprehensive benchmark to measure language models' abilities to capture underlying social information. Motivated by theories of systemic functional linguistics and interpersonal pragmatics, SOCKET cuts across a number of dimensions of interpersonal communication, allowing it to also be a tool to better understand and interpret co-learning abilities and dependencies in sociolinguistic tasks. Having this ability allows researchers and users to more efficiently and effectively deploy NLP methods by providing empirical results on the limits and affordances of a variety of out-of-domain social language tasks.

In total, SOCKET spans 2,616,342 items across all tasks, including 269,246 samples in the test set. However, experimenting with an evaluation set of size can be prohibitive due to model size, available resources, and considerations of the environment. Therefore, we also release a subset of our data as SOCKETTE (SOCKET but **T**ini**e**r) that contains at most 1000 items per task in the test set, reducing the test set to 43,731 samples. In Appendix B.3, we show that performance on SOCKETTE is highly correlated and we hope that this smaller subset enables more rapid progress.

## 4 Benchmarks on the Social Knowledge Capabilities of LLMs

We first train and evaluate several commonly used multitask LLMs on our datasets to obtain benchmark results, which provide a first glimpse of how good LLMs are at learning social knowledge tasks. Experiment details are described in Appendix §B.

### 4.1 Training Methods

**BERT-based Finetuning** We first apply the standard process of fine-tuning on pretrained LLMs. We select two of the most popular LLMs - BERT (Devlin et al., 2019) and RoBERTa (Liu et al., 2019) - as well as two lightweight models known to achieve high performance on finetuning tasks - DeBERTa-V3 (He et al., 2021) and MiniLM (Wang et al., 2020).

**Prompt-based finetuning** Prompt-based finetuning has emerged as a flexible and effective means of adapting models to downstream tasks (Wei et al., 2021). As a benchmark, we include the performances of a T5 model (Raffel et al., 2020) trained on each task via finetuning. We manually design prompts for each task. For classification tasks, we use verbalizers to map the class to word labels and for regression tasks, we adopt a method similar to Gao et al. (2021) in that we use two anchor words "Yes" and "No" and consider the probability of predicting "Yes" as the final score. For span-based tasks, we train the model to directly generate the sequence outputs. A list of prompts can be found in Table 8 and Table 9 in the Appendix.

**Zero-shot predictions** We further apply our designed prompts to test the performances of LLMs in a zero-shot setting where no further finetuning is performed. Using the same prompts proposed in Table 8, we test SOCKET on several widely used LLMs: GPT (Radford et al., 2018), GPT-J-6B (Wang and Komatsuzaki, 2021), OPT

| Category | Model | No. params (B) | Humor & Sarcasm | Offens. | Sent. & Emo. | Social Factors | Trust. | CLS | PAIR | REG | SPAN | *Avg.* |
|---|---|---|---|---|---|---|---|---|---|---|---|---|
| baseline | majority | | 0.27 | 0.42 | 0.12 | 0.25 | 0.41 | 0.39 | 0.34 | 0.50 | 0.00 | 0.32 |
| | random | | 0.40 | 0.35 | 0.17 | 0.36 | 0.35 | 0.38 | 0.51 | 0.50 | 0.00 | 0.32 |
| zero-shot | EleutherAI-gpt-j-6b | 6 | 0.39 | 0.35 | 0.29 | 0.33 | 0.28 | 0.32 | 0.26 | 0.50 | 0.08 | 0.32 |
| | alpaca-native | 7 | 0.39 | 0.44 | 0.45 | 0.55 | 0.31 | 0.42 | 0.48 | 0.57 | 0.17 | 0.43 |
| | bigscience-bloomz-7b1 | 7 | 0.50 | 0.49 | 0.43 | 0.53 | 0.45 | 0.49 | 0.51 | 0.56 | 0.09 | 0.48 |
| | cerebras-Cerebras-GPT-6.7B | 6.7 | 0.42 | 0.39 | 0.30 | 0.36 | 0.34 | 0.35 | 0.33 | 0.52 | 0.13 | 0.36 |
| | decapoda-research-llama-13b | 13 | 0.49 | 0.43 | 0.36 | 0.42 | 0.31 | 0.38 | 0.52 | 0.53 | 0.17 | 0.40 |
| | facebook-opt-13b | 13 | 0.31 | 0.40 | 0.19 | 0.22 | 0.28 | 0.31 | 0.25 | 0.49 | 0.15 | 0.31 |
| | google-flan-t5-xxl | 11 | 0.66 | 0.56 | 0.52 | 0.60 | 0.49 | 0.56 | 0.64 | 0.63 | 0.17 | 0.55 |
| | t5-3b | 3 | 0.34 | 0.41 | 0.27 | 0.32 | 0.35 | 0.36 | 0.36 | 0.49 | 0.13 | 0.36 |
| | llama2-7b-chat | 7 | 0.39 | 0.27 | 0.33 | 0.34 | 0.24 | 0.25 | 0.38 | 0.56 | 0.18 | 0.29 |
| | GPT-3.5 | | 0.64 | 0.55 | 0.57 | 0.65 | 0.45 | 0.57 | 0.49 | 0.67 | 0.21 | 0.56 |
| finetuning | bert-base-uncased | 0.11 | 0.78 | 0.76 | 0.65 | 0.70 | 0.62 | 0.70 | 0.79 | 0.77 | 0.55 | 0.71 |
| | roberta-base | 0.086 | 0.79 | 0.77 | 0.68 | 0.72 | 0.63 | 0.70 | 0.83 | 0.79 | **0.64** | 0.72 |
| | deberta-v3 | 0.098 | **0.83** | **0.77** | **0.70** | **0.72** | **0.66** | **0.72** | **0.87** | **0.79** | 0.63 | **0.73** |
| | MiniLM | 0.066 | 0.77 | 0.72 | 0.61 | 0.67 | 0.58 | 0.66 | 0.78 | 0.69 | 0.57 | 0.67 |
| | T5* | 0.25 | 0.68 | 0.72 | 0.55 | 0.59 | 0.47 | 0.65 | 0.66 | 0.45 | 0.54 | 0.62 |

Table 2: A comparison of the benchmark performances of different models and training schemes. Best-performing instances are shown in bold. The best performing parameter size for each zero-shot model is shown (cf. Figure 1) . A full comparison of all models across all settings can be found in Table 4 in the Appendix. The performances on each individual task using a DeBERTa-V3 model can be found in Table 10 in the Appendix.

(Zhang et al., 2022), T5 (Raffel et al., 2020), LLaMA (Touvron et al., 2023a), LLaMA-2 (Touvron et al., 2023b), BLOOM (Workshop et al., 2023), BLOOMZ (Muennighoff et al., 2022), FLAN-T5 (Chung et al., 2022b), RedPajama (Computer, 2023), and Alpaca (Taori et al., 2023; Wang et al., 2022b). We also evaluate the performance of GPT-3.5 [3] using OpenAI's API. Samples for which a model does not provide an appropriate label are automatically marked as incorrect. For each LLM variant, we test zero-shot results for different model sizes ranging between 110M and 13B parameters, which we report in Table 4 in the Appendix.

## 4.2 Results

We compare model performances across category type and task type as shown in Table 2. Each reported value is the average of the scores on every task within the specified group. The rationale behind using a unified average score is to provide a high-level comparison of the performances of zero-shot and fine-tuned models under various settings, including task type (regression/classification/pair/span) as well as dimension of social knowledge.

DeBERTa-V3 achieves the best overall performance after full training on each of the SOCKET datasets, followed by other BERT-based models. The prompt-based finetuning of T5 performs worse than standard finetuning, especially on the pairwise classification and regression tasks. Meanwhile, most zero-shot models perform only slightly better than the baseline, indicating that prompting alone does not elicit correct social knowledge—though

[3]https://platform.openai.com/docs/models/gpt-3-5

two models, google-flan-t5-xxl and GPT3.5, are much closer in performance to supervised models.

**Social knowledge can be hard to infer** Our benchmark results reveal that even our best-performing model leaves significant room for improvement, scoring just above 0.7 overall—compared with the models' analogous performance on syntactic and discourse NLU tasks (He et al., 2021) which are often much higher. A comparison among categories of social knowledge reveals that humor & sarcasm is generally the easiest to detect, while trustworthiness is the hardest. This performance gap can be attributed to the level of understanding required for each dimension - while detecting humor or other social emotions can often be correlated with cues such as sentiment, detecting the level of trust within sentences requires more understanding of the context and may be harder to detect using computational models (Choi et al., 2020). At a task level, we observe that models struggle most in span detection tasks. This is a complex task due to its open-ended nature, and thus BERT-based finetuning does not perform as well as in other types of tasks. We highlight that learning the various aspects of social knowledge is indeed a challenge for current LLMs, and thus call for the need for future models with improved social capabilities.

**Supervised models significantly outperform zero-shot models** Table 2 reveals that despite being much smaller in the number of parameters, finetuning supervised models such as MiniLM leads to much better performance than zero-shot models using state-of-the-art LLMs. All the zero-shot LLMs performed poorly, many on par with random baselines, apart from FLAN-T5. Figure 1 shows

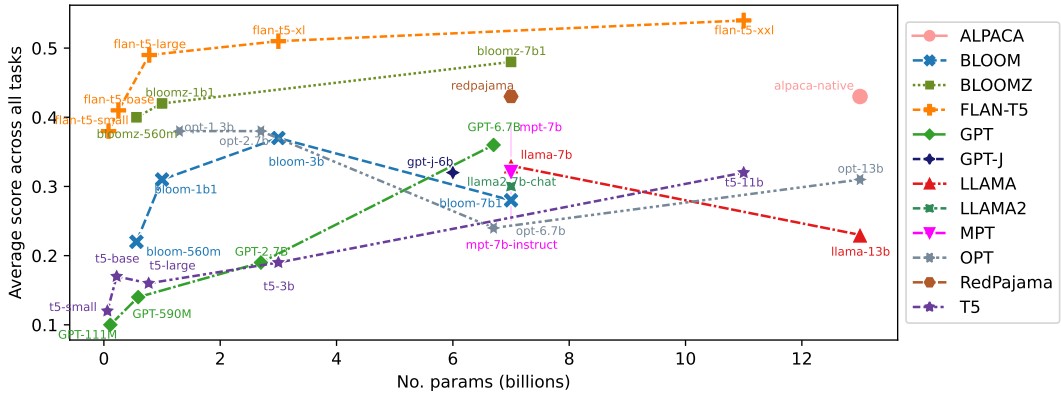

Figure 1: A comparison of LLMs on the aggregated scores tested on SOCKET under zero-shot settings. The overall performances vary greatly by model architecture, while larger models do not always guarantee better performance.

a detailed picture of how different LLM parameter sizes influence the ability to comprehend social knowledge tasks in a zero-shot setting. Surprisingly, we find that of the various training schemes FLAN-T5 is by far the most effective for inferring social knowledge, even with relatively small models. We speculate this performance is due to its initial pretraining on more than 1,000 tasks.

**More parameters do not guarantee more social knowledge** Another general trend we observe is a weak correlation between the number of parameters and overall performance within the same model architecture ($\rho = 0.266$, $p = .08$). This is to some extent determined by the model's ability to understand the task itself given an instruction prompt as well as a sample input, as larger models are capable of understanding a wider variety of tasks (cf. Appendix Table 6). Of course, it is also possible that larger LLMs could encode a greater amount of social knowledge through their greater parameter sizes. Interestingly, we observe that for some models, larger size does not always guarantee better performance. This is the case especially for BLOOM, T5 and GPT, where the largest model is not always the best performer within the group.

Models varied in the ability to follow instructions (Appendix Table 6). As expected, instruction-tuned models like FLAN-T5 and Alpaca are generally able to follow the prompt instructions, while other models may generate answers that are not provided in the options. For our social tasks, instruction-following was not significantly correlated with model size ($\rho$=0.08, p=0.60). Thus, lower model performance in Figure 1 is, in part, due to models being unable to answer questions relating to social knowledge.

**When models are able to answer the question, are they right?** Restricting only to instances in which a model outputs a valid answer reveals heterogeneity among different model groups (Figure 3), showing an interplay between model size, coverage, and performance. For architectures such as FLAN-T5 or BLOOMZ we observe a positive correlation between parameter size and performance, both in its ability to understand instructions and to make correct predictions. On the other hand, for certain architectures having larger parameters can actually make it worse at understanding instructions (e.g. LlaMA) or predicting correctly (e.g. OPT). Recognizing that measuring of instruction understanding and the accuracy of an LLM both depend on how strictly one chooses to map the predictions to an answer, overall, our results suggest that while LLMs do contain the potential for understanding social knowledge, additional steps such as finetuning or instruction tuning are likely needed for better social understanding.

## 5 Do we see Cross-task Transfer of Social Knowledge?

In this section, we examine the relations and dependencies between tasks using the predictions of LLMs trained on different tasks and test for dependencies between tasks that are predicted by theory.

**Quantifying Task Dependency** We quantify the dependency between two tasks as follows. We finetune a pretrained LLM on task $t_A$ to obtain a model $m_A$, which is used to make predictions on the test set of another task $t_B$. The correlation between the predicted values from model $m_A$ and the true labels of the test set of $t_B$ is considered as the task

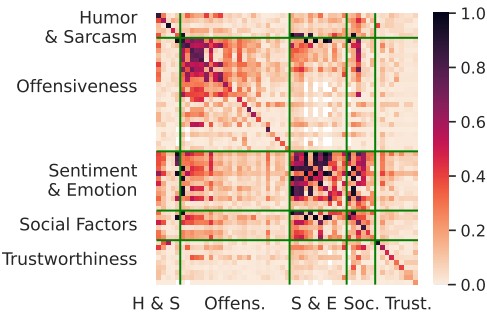

Figure 2: Heatmap of task dependency among all task pairs, annotated at category level. Each value represents the absolute strength of correlation between the true labels of the test set of a specific task (columns) and the predictions made on that task using a model trained on a different task (rows). We observe strong correlations, especially within the Offensiveness, Sentiment & Emotion, and Social Factors categories. A larger version labeled at the task level is shown in Appendix Figure 6.

dependency that $t_A$ has on $t_B$. We report the absolute correlation value, as negatively correlated tasks are still informative. We describe how the correlations are obtained across different task types in the Appendix (§B.6). Span identification tasks are omitted from this analysis, resulting in $55 \times 55$ scores. We also measure the pairwise correlation between models $m_A$ and $m_B$ as well as task dependency to gain an additional perspective of task similarity. Details for the model correlation can be found in Appendix §B.6 and Figure 7.

The task dependencies for all task pairs, shown in Figure 2, reveal salient block structures within the category,[4] especially for the Offensiveness, Sentiment & Emotion, and Social Factors categories, suggesting the existence of shared knowledge within our thematically grouped tasks. These correlations align with existing findings from interpersonal pragmatics on the relationships between social knowledge. For instance, increased self-disclosure or pain-related interactions are known to promote both intimacy (*questionintimacy*) and empathy (*empathy*) (Parks, 1981; Cano and Williams, 2010), two elements within the Social Factors category, while the usage of emojis (*tweet_emoji*) as effective symbols are indicative of emotional states such as valence (*emobank#_valence*) and arousal (*emobank#_arousal*) (Fischer and Herbert, 2021), which belong to the Sentiment & Emotion category.

The Offensiveness category shows mixed results in comparison with Arango et al. (2019), whose

results show that hate speech datasets are often overfit and do not generalize well to other similar datasets . Figures 2 & 6, however, show that of the seven datasets included in SOCKET, five of them included at least one task which showed comparable correlations when tested both within and out of domain. Indeed, *PersonDirectedAbuse*, a task labeled for offensive language specifically directed towards an individual, is actually predicted better by models fine-tuned on jigsaw# tasks than it was on its own.

Interestingly, correlations are scarce within the Humor & Sarcasm, and Trustworthiness categories. This is consistent with findings from (Hu et al., 2022b) which show that models without exposure to linguistic forms lack the requisite social information to perform well on non-literal pragmatic phenomena such as humor and deceit.

Another notable individual task is *humor_rating* from the Humor & Sarcasm dataset, which performs well as both the fine-tuning and predicted task alongside a number of tasks from the Emotion & Sentiment category—particularly discretized emotion tasks, as well as *hateoffensive* in the Offensiveness category—which labels comments as either "hateful," "offensive," or neither. While relationships between offensiveness and humor have been theorized as early as Freud (1960) and sentiment recognition has been shown to bolster offensive language detection (Liu, 2012), relatively little has been said regarding connections between the three categories and thus, this result presents an opportunity for further research.

We observe that *politeness* shows strong transfer with many of the offensive and hate speech detection tasks in the SOCKET benchmark. In particular, those tasks with high correlation within the offensive category are highly correlated in predicting the politeness classification task. This finding is supported by literature showing that impoliteness can fall under the umbrella of offensive language (Bączkowska, 2021) and, although key differences exist in the pragmatics of the two, the constructs are closely related (Parvaresh, 2023; Culpeper, 2021).

Interestingly, regression tasks (from the *hahackathon*, *emobank*, and *empathy* datasets) in general have strong correlations with several other tasks. This trend suggests that tasks labeled with continuous variables may have more expressive power compared to ordinal or nominal categorization, and thus have a higher potential for stronger

---

[4]See Figure 6 for fully labeled version.

task dependencies. However, the magnitude of the correlation may be influenced by the relative value distributions of different correlation methods. This finding calls for a need for more datasets with continuous labels, which requires more effort but allows models to capture more fine-grained concepts of social knowledge.

# 6 Can Multi-task Training improve Social Knowledge?

Our findings reveal significant task transfer, both within and across task categories, which hints at shared knowledge among tasks. Linguistics studies of social language also note the interrelated perceptions of different dimensions such as humor and offensiveness (Culpeper, 2021; Attardo, 2008; Alberts, 1992; LI Hai-hui, 2019). We now examine whether LLMs can learn a more robust sense of social knowledge by training on multiple tasks.

**Experimental Setup** Recent studies have explored the possibility of multi-task training on LLMs, which is training a single model on several different tasks simultaneously, with effects of improving its performance on both seen and unseen tasks (Aghajanyan et al., 2021; Padmakumar et al., 2022). We apply multi-task training on SOCKET, but make one clear distinction from prior work. Whereas previous studies have shown that multi-task training is especially effective when the grouped tasks are of similar types (Padmakumar et al., 2022), we introduce a new setting by grouping tasks instead by our defined categories of social knowledge. We expect that same-category tasks contain social knowledge that can be shared across tasks, resulting in LLMs that learn a more robust concept of the specific dimension than when trained on single tasks.

A popular method for multi-task training is pre-finetuning (Aghajanyan et al., 2021; Shi et al., 2022), which involves a first stage of finetuning on multiple tasks using task-specific heads on a shared encoder, then re-using the encoder for downstream tasks. We apply pre-finetuning in two different settings: (1) *category-wise tasks*, where we perform pre-finetuning on tasks grouped to the same category, and (2) *all tasks*, where all tasks of SOCKET are included in the pre-finetuning stage. Consistent with prior work, we perform the second finetuning stage on individual tasks using the pre-finetuned model as initial weights (Aghajanyan et al., 2021). Other training details are identical to §4.

**Results** Multitask training had little to negative ef-

| | Model type | | |
|---|---|---|---|
| Category | Single task | Category-wise | All tasks |
| Humor & Sarcasm | 0.76 | 0.76 | 0.74* |
| Offensiveness | 0.76 | 0.76 | 0.76 |
| Sentiment & Emotion | 0.64 | 0.64 | 0.62 |
| Social Factors | 0.67 | 0.67 | 0.66 |
| Trustworthiness | 0.66 | 0.64* | 0.62* |

Table 3: The performances of different multi-task settings aggregated at category level. Numbers with * indicate cases where the prediction results significantly differ from the single task setting (paired t-tests).

fect on task performance (Table 3). Although some tasks did benefit from being co-trained within category (Appendix Table 10)—particularly the Offensiveness category—when aggregated at the category level, the average performance is worse. In particular, the Humor & Sarcasm and Trustworthiness categories have the lowest levels of within-task and cross-task dependencies (§5). The performance drop is less strong in categories with high dependency, indicating that while multi-task training on similar tasks may not always improve performance, task-relatedness can help preserve performance when also learning task-specific new concepts. Together, our results suggest multi-task training on unrelated social tasks hurts overall performance—a result contrary to social science expectations of how social information is processed—and points to a need to further investigate cases when applying multi-task training as a practice to improve the social knowledge of LLMs.

# 7 Conclusion

People increasingly interact with LLMs in natural conversation. To what degree are these models able to pick up on the social cues? To help answer this question, we introduce SOCKET, an NLP benchmark to evaluate how well models perform at learning and recognizing concepts of social knowledge. We provide benchmark results using several popular models and provide case studies of studying the inherent social capabilities of LLMs in a zero-shot setting. Surprisingly, LLMs perform moderately at best, with even large LLMs (>10b parameters) varying widely in their abilities. Additionally, we show that there exist significant task dependencies both within and across task categories, and that multi-task training on task categories can affect model performance. Our work contributes to the broader NLP community by fostering future efforts toward building and evaluating more socially responsible and coherent LLMs.

## 8   Limitations

**Cross-cultural and multilingual expansions**
Culture is an important aspects of understanding language, especially within the broader setting of multilingual NLP. In this study, however, we make a clear distinction between cultural knowledge and social knowledge, the latter of which is our focus for this study. Our work is grounded in social-psychological theory and the sociolinguistics of interpersonal communication, especially dyadic communication. Such studies are often aimed at phenomena that are widely shared across cultures while recognizing that cultural variation exists within how those phenomena are perceived. In contrast, work in anthropology or cultural studies provides a different perspective and grounding. Such work frequently focuses on cross-cultural perspectives and what is or is-not shared across cultures. For example, in language, the interpretation of whether something is polite can depend on gender norms (Mills, 2004) and cultural (Lorenzo-Dus and Bou-Franch, 2003), highlighting the potential context sensitivity. Similarly, the perception of toxicity can depend on the cultural identities of the reader (Sap et al., 2019a; Ghosh et al., 2021). While highly valuable to study, cultural knowledge is a separate construct from social knowledge (though interrelated) and not the focus of this benchmark, though we hope that our work inspires other benchmarks to help assess such differences.

Regarding multilingual data, SOCKET currently contains tasks based in English due to the limited availability of tasks in non-English. While there are a few datasets such as HAHA (Chiruzzo et al., 2020) in Spanish and DeTox (Demus et al., 2022) in German, we were not able to find sufficient numbers yet to provide a meaningful grouping. This highlights the importance of constructing datasets and frameworks capable of capturing social knowledge for a wide variety of languages, which we consider an important future step.

**Additional dimensions and forms of social knowledge** Interpersonal communication conveys a richness of different social information and despite our extensive literature review and data curation process, we fully acknowledge that other dimensions of social knowledge are not included in our current benchmark. In creating SOCKET, our aim was to focus on diverse categories of social knowledge that have multiple tasks in order to

get a more robust assessment of model capabilities, e.g., multiple tests of a model's ability to recognize humor, in order to avoid the pitfalls of ascribing progress on the basis of a single task alone (Subramonian et al., 2023). Nevertheless, SOCKET omits several notable dimensions or forms of social knowledge. Some social aspects of language such as pragmatic polysemy (Carston, 2021; Apresjan, 1974) and idioms (Strässler, 1982) either had too few similar datasets to form a theory-backed category, or there were no existing NLP datasets to test the construct. The latter is the case, especially in the case of linguistic techniques unique to recognize when a speaker is adopting community-specific dialects such as African-American English (Hyter et al., 2015; Rivers et al., 2012; Allan, 2007) and Queer Language (Barrett, 2006; Huebner, 2021; Harvey, 2000).

Social language understanding happens within a static, unspecified context for the current tasks in SOCKET. However, the social context in which a message is said can dramatically alter its meaning. NLP is just beginning to incorporate the social context into language understanding (Hovy and Yang, 2021). While a handful of datasets have begun to explore modeling context explicitly, such as through the preceding conversation (Pavlopoulos et al., 2020; Menini et al., 2021), the identity of the speaker (Almagro et al., 2022), the social relationship between speakers (Jurgens et al., 2023), or explicit social norms (Park et al., 2021), there are currently too few of such tasks to compose a comprehensive benchmark with which to measure progress. Future datasets and benchmarks will be needed to study understanding social knowledge when controlling for context.

Thus, SOCKET represents a starting point for modeling models' abilities and provides room for improvement via the addition of new categories or constructs, as additional data becomes available. Further inclusion of other dimensions and corresponding tasks should be an ongoing goal.

**Benchmarks as markers of progress** SOCKET fills a current gap for assessing the capabilities of LLMs on understanding social language. However, benchmarks as constructs have been rightly critiqued as markers of progress in NLP (e.g., Bowman and Dahl, 2021; Schlangen, 2021; Subramonian et al., 2023), due to aspects such as changing or narrowing the field's definition of a task, overemphasizing or overselling progress in a particular

area, or encouraging leaderboard chasing. In designing SOCKET, we aimed to directly address the pitfalls of benchmark design by selecting a diverse set of social language understanding tasks that mirrored human capabilities recognized in social science studies; this selection helps ensure a broad measure of performance and that "progress" is not due to improved performance on one type of task. However, the benchmark itself does not capture all of social knowledge (nor do we claim as such) and we view it only as a starting point—a yardstick by which to measure current systems—with a need for new tasks and benchmarks as models advance in their social reasoning capabilities.

The use of a single metric to measure progress in an area or task can mask meaningful insight and fail to contextualize performance. While we follow common practice in NLP (e.g., Wang et al., 2018, 2019; Muennighoff et al., 2023) and report a single mean score in Table 2, the design of SOCKET includes specific task categories and types designed to easily and meaningfully inspect what is ultimately contributing to the single score—e.g., are models performing well in classification but poorly in span recognition? Nevertheless, this design is a trade-off: A single score can and likely does promote leaderboard chasing by setting a clear goal to pursue, while completely disaggregated scores like those in Table 4 become unwieldy and make it hard to assess whether meaningful progress is being made when comparing two models. Here, we have opted to report both the overall average and averages for each category and type (10 scores total) in an attempt to balance these two tensions.

**Technical limitations** One major limitation of the current benchmark is we only tested LLMs that have up to 13B parameters. Recent studies show that the LLMs may start to show emergent abilities when they are scaled up above a certain threshold (Wei et al., 2022). Due to limited computational and financial resources, we are not able to test all very large language models, though we welcome future researchers to work on our benchmark and evaluate the sociability of new and larger LLMs.

Finally, our zero-shot model performance used curated prompts on pretrained models without any further finetuning. While it is widely known that instruction-based finetuning specific to downstream tasks can greatly improve performance, we deliberately chose not to do so. Finetuning LLMs with billions of parameters can leave a large carbon foot-print, which we avoid for both financial and environmental reasons (Hu et al., 2021; Liu et al., 2022; Lester et al., 2021).

## 9 Ethical Considerations

The interpretation of social information in communication is highly subjective in that it can largely vary depending on demographic and contextual factors. Nevertheless, several NLP datasets are created via crowdsourcing, which raises concerns on whether the dataset's labels are truly representative of our society (Talat et al., 2022). Even within our benchmark, there is the possibility that for tasks such as offensiveness or humor the crowdsourced labels may undermine phrases that might disregard a specific demographic group, which may be inevitably picked up by LLMs that are trained and evaluated on these datasets. Improved versions of our benchmark should include datasets that are more inclusive in such contexts, which we call for future work.

There has been increasing concern over the amount of computing resources required for conducting deep learning research at scale, especially regarding LLMs where task performance is improved through larger datasets, increased model parameters, and longer training hours. The time and amount of computing resources required for training LLMs has become nontrivial (Bender et al., 2021), and it has been increasingly aware among machine learning practitioners to consider the carbon footprint of models and computing methods to minimize risks of global warming. This, combined with limited transparency of experiment results, may harm the very concept of open science. Keeping this in mind, we focused on conducting easily reproducible experiments that can be run on a single GPU within the time frame of hours or a couple of days at the longest. Some of our findings contribute towards this rightful direction, as can be seen in our investigation on multi-task training.

More importantly, we highlight the fact that the main contribution of our study is a thoroughly designed public framework of tasks for examining the social knowledge of LLMs. While it is indeed important to develop and improve LLMs that can perform better on several tasks, we believe that correctly evaluating the level of social knowledge engraved in these models is an equally important task. Without such scrutiny, the users of LLMs deployed in practical settings may be vulnerable to

socially undesirable or unethical content. We sincerely hope that our efforts in producing SOCKET can ease difficulties of conducting future studies that aim to examine and improve the social understanding of LLMs.

## Acknowledgments

The authors thank reviewers for their timely and valuable feedback on the paper, with a special shout-out to R1 for their very detailed feedback which certainly made this paper better. We also thank the members of the Center for Social Media Responsibility, especially Paul Resnick and James Park for their support which enabled the initiation of this project. This work was supported by the National Science Foundation under Grant Nos. IIS-2007251, IIS-2143529, and 2137469. The third author was partially supported by grant SES-2200228 from the National Science Foundation.

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

## A   Details on dataset processing

### A.1   Benchmark construction (§3)

The SOCKET dataset consists of 58 tasks from 35 unique, public datasets. The datasets that make up the benchmark dataset are processed in a way that is meant to balance uniformity across datasets and tasks while minimizing deviations from the original dataset.

For all datasets, key changes from the original dataset are twofold:

- Duplicates and unlabeled items are removed from all datasets. If duplicates occur across data splits, the splits are recombined, reshuffled, and split.

- All datasets are split 80%/10%/10% between train/test/dev splits, respectively. Any datasets not split 80%/10%/10% are recombined, reshuffled, and split 80%/10%/10%.

All datasets were made compatible with the Hugging Face Datasets package.

## B   Experimental Details

### B.1   Computational resources (§4, §5, §6)

All of our experiments were conducted on an Ubuntu 22.04.1 machine installed with NVIDIA RTX A5000 and A6000 GPUs. The Python packages used in our experiments include Pytorch 1.13, Transformers 4.21.3, and Pytorch Lightning 1.6.4.

### B.2   Comparison of all models

Table 4 contains a detailed version of Table 2, where the scores of every single task are presented.

### B.3   Details on the comparison between SOCKET and SOCKETTE

32 out of 58 tasks contained more than 1,000 test samples, resulting in a disparity between the sizes of the original SOCKET and SOCKETTE variants. To test that both datasets still offer comparable evaluations for testing models, we compare their scores for a supervised model and compare test set performances. For each task, we train a deberta-v3-base model, evaluate using the test sets of both versions, and compute the correlation between each setting using Pearson's r score. We provide evaluation results of SOCKETTE for our models in Table 5. Also, we show through Table 7 and Figure 5 that there exists a strong correlation between

the evaluations of both versions, demonstrating that SOCKETTE is indeed a representative sample of SOCKET.

### B.4   Details on language model finetuning (§4, §5, §6)

#### B.4.1   Task-specific heads (§4, §5, §6)

As our benchmark consists of four different task types: classification, regression, sentence pair detection, and span identification - we maintain a unified structure for each task where each sample is fed into the encoder of an LLM, and the output states are then fed into a task-specific head layer. For span detection tasks, we feed the last hidden layer into a bidirectional GRU (Chung et al., 2014), and then the output vectors of the GRU into a linear layer that transforms each vector into a dimension of 3, corresponding to the [B,I,O] labels for each token, following earlier work in span identification (Suman and Jain, 2021). For all other tasks, we feed the last hidden state of the encoder corresponding to the [CLS] token into a separate classifier/regression head consisting of two linear layers of hidden size 768 and a dropout probability of 0.1. We use the mean squared error loss for regression tasks and the cross-entropy loss for all other tasks.

#### B.4.2   Training strategies for language model finetuning (§4, §6)

When training models for the benchmark (§4) and the multi-task (§6) experiments, the learning rate was linearly increased for 6% of the training steps up to 1e-5 and linearly decreased afterward. All models were trained for a maximum of 10 epochs using three different seeds, with early stopping after validation performance did not increase for three consecutive epochs.

Our multi-task training in §6 requires two stages of training: (1) a pre-finetuning stage that simultaneously trains a model on multiple different tasks, and (2) a finetuning stage that loads the model trained from (1) and finetunes it to a single task. In the first stage, a single batch can include several different tasks and produce different types of losses. To obtain a unified loss that is differentiable, we aggregated the loss for each sample and sum them up, which we use for backpropagation. For both stages, we use the same aforementioned training steps and learning rate strategy.

For all settings, the training batch size was set to 32 with 16-bit precision enabled. Validation was

| Category | Model | No. params (billions) | Humor & Sarcasm | Offensiveness | Sentiment & Emotion | Social Factors | Trustworthiness | CLS | PAIR | REG | SPAN | *Avg.* |
|---|---|---|---|---|---|---|---|---|---|---|---|---|
| baseline | majority | | 0.27 | 0.42 | 0.12 | 0.25 | 0.41 | 0.39 | 0.34 | 0.50 | 0.00 | 0.32 |
| | random | | 0.40 | 0.35 | 0.17 | 0.36 | 0.35 | 0.38 | 0.51 | 0.50 | 0.00 | 0.32 |
| zero-shot | EleutherAI-gpt-j-6b | 6 | 0.39 | 0.35 | 0.29 | 0.33 | 0.28 | 0.32 | 0.26 | 0.50 | 0.08 | 0.32 |
| | alpaca-native | 7 | 0.39 | 0.44 | 0.45 | 0.55 | 0.31 | 0.42 | 0.48 | 0.57 | 0.17 | 0.43 |
| | bigscience-bloom-560m | 0.56 | 0.30 | 0.24 | 0.14 | 0.26 | 0.18 | 0.21 | 0.26 | 0.49 | 0.07 | 0.22 |
| | bigscience-bloom-1b1 | 1 | 0.26 | 0.39 | 0.17 | 0.25 | 0.33 | 0.33 | 0.21 | 0.48 | 0.11 | 0.31 |
| | bigscience-bloom-3b | 3 | 0.37 | 0.44 | 0.30 | 0.34 | 0.30 | 0.37 | 0.32 | 0.51 | 0.12 | 0.37 |
| | bigscience-bloom-7b1 | 7 | 0.42 | 0.27 | 0.30 | 0.29 | 0.20 | 0.25 | 0.39 | 0.51 | 0.11 | 0.28 |
| | bigscience-bloomz-560m | 0.56 | 0.38 | 0.42 | 0.34 | 0.42 | 0.41 | 0.40 | 0.41 | 0.51 | 0.10 | 0.40 |
| | bigscience-bloomz-1b1 | 1 | 0.41 | 0.44 | 0.38 | 0.43 | 0.41 | 0.42 | 0.44 | 0.52 | 0.10 | 0.42 |
| | bigscience-bloomz-7b1 | 7 | 0.50 | 0.49 | 0.43 | 0.53 | 0.45 | 0.49 | 0.51 | 0.56 | 0.09 | 0.48 |
| | cerebras-Cerebras-GPT-111M | 0.11 | 0.21 | 0.07 | 0.18 | 0.09 | 0.03 | 0.05 | 0.18 | 0.49 | 0.07 | 0.10 |
| | cerebras-Cerebras-GPT-590M | 0.59 | 0.31 | 0.11 | 0.14 | 0.19 | 0.09 | 0.13 | 0.23 | 0.00 | 0.08 | 0.14 |
| | cerebras-Cerebras-GPT-2.7B | 2.7 | 0.32 | 0.19 | 0.14 | 0.23 | 0.16 | 0.17 | 0.26 | 0.49 | 0.18 | 0.19 |
| | cerebras-Cerebras-GPT-6.7B | 6.7 | 0.42 | 0.39 | 0.30 | 0.36 | 0.34 | 0.35 | 0.33 | 0.52 | 0.13 | 0.36 |
| | decapoda-research-llama-7b | 7 | 0.45 | 0.44 | 0.34 | 0.41 | 0.22 | 0.38 | 0.34 | 0.52 | 0.12 | 0.38 |
| | decapoda-research-llama-13b | 13 | 0.49 | 0.43 | 0.36 | 0.42 | 0.31 | 0.38 | 0.52 | 0.53 | 0.17 | 0.40 |
| | facebook-opt-1.3b | 1.3 | 0.41 | 0.43 | 0.28 | 0.33 | 0.39 | 0.38 | 0.34 | 0.50 | 0.12 | 0.38 |
| | facebook-opt-2.7b | 2.7 | 0.37 | 0.45 | 0.29 | 0.36 | 0.32 | 0.37 | 0.34 | 0.52 | 0.18 | 0.38 |
| | facebook-opt-6.7b | 6.7 | 0.20 | 0.31 | 0.20 | 0.16 | 0.17 | 0.21 | 0.13 | 0.48 | 0.16 | 0.24 |
| | facebook-opt-13b | 13 | 0.31 | 0.40 | 0.19 | 0.22 | 0.28 | 0.31 | 0.25 | 0.49 | 0.13 | 0.31 |
| | google-flan-t5-small | 0.08 | 0.37 | 0.43 | 0.23 | 0.32 | 0.40 | 0.38 | 0.34 | 0.50 | 0.08 | 0.37 |
| | google-flan-t5-base | 0.25 | 0.45 | 0.49 | 0.43 | 0.45 | 0.41 | 0.47 | 0.44 | 0.53 | 0.09 | 0.45 |
| | google-flan-t5-large | 0.78 | 0.50 | 0.52 | 0.48 | 0.50 | 0.41 | 0.50 | 0.44 | 0.59 | 0.13 | 0.49 |
| | google-flan-t5-xl | 3 | 0.59 | 0.53 | 0.50 | 0.60 | 0.47 | 0.53 | 0.58 | 0.60 | 0.19 | 0.52 |
| | google-flan-t5-xxl | 11 | 0.66 | 0.56 | 0.52 | 0.60 | 0.49 | 0.56 | 0.64 | 0.63 | 0.17 | 0.55 |
| | mosaicml-mpt-7b | 7 | 0.41 | 0.42 | 0.36 | 0.44 | 0.32 | 0.37 | 0.45 | 0.54 | 0.25 | 0.39 |
| | mosaicml-mpt-7b-instruct | 7 | 0.20 | 0.25 | 0.30 | 0.34 | 0.15 | 0.22 | 0.46 | 0.24 | 0.25 |
| | t5-small | 0.06 | 0.12 | 0.05 | 0.09 | 0.18 | 0.06 | 0.08 | 0.09 | NaN | 0.04 | 0.08 |
| | t5-base | 0.22 | 0.41 | 0.18 | 0.15 | 0.31 | 0.12 | 0.20 | 0.30 | NaN | 0.03 | 0.20 |
| | t5-large | 0.77 | 0.36 | 0.11 | 0.26 | 0.26 | 0.10 | 0.14 | 0.30 | 0.50 | 0.03 | 0.18 |
| | t5-3b | 3 | 0.34 | 0.41 | 0.27 | 0.32 | 0.35 | 0.36 | 0.36 | 0.49 | 0.13 | 0.36 |
| | t5-11b | 11 | 0.38 | 0.38 | 0.14 | 0.31 | 0.36 | 0.33 | 0.23 | 0.50 | 0.03 | 0.32 |
| | togethercomputer-RedPajama-INCITE-7B-Instruct | 7 | 0.41 | 0.48 | 0.37 | 0.41 | 0.42 | 0.44 | 0.39 | 0.53 | 0.11 | 0.43 |
| | llama2-7b-chat | 7 | 0.39 | 0.27 | 0.33 | 0.34 | 0.24 | 0.25 | 0.38 | 0.56 | 0.18 | 0.29 |
| | GPT-3.5 | | 0.64 | 0.55 | 0.57 | 0.65 | 0.45 | 0.57 | 0.49 | 0.67 | 0.21 | 0.56 |
| full | bert-base-uncased | 0.11 | 0.78 | 0.76 | 0.65 | 0.70 | 0.62 | 0.70 | 0.79 | 0.77 | 0.55 | 0.71 |
| | roberta-base | 0.086 | 0.79 | 0.77 | 0.68 | 0.72 | 0.63 | 0.70 | 0.83 | 0.79 | **0.64** | 0.72 |
| | deberta-v3 | 0.098 | **0.83** | **0.77** | **0.70** | **0.72** | **0.66** | **0.72** | **0.87** | **0.79** | 0.63 | **0.73** |
| | MiniLM | 0.066 | 0.77 | 0.72 | 0.61 | 0.67 | 0.58 | 0.66 | 0.78 | 0.69 | 0.57 | 0.67 |
| | T5* | 0.25 | 0.53 | 0.71 | 0.48 | 0.50 | 0.47 | 0.63 | 0.44 | 0.37 | 0.54 | 0.58 |

Table 4: A comparison of the benchmark performances of different models and training schemes. Best-performing instances are shown in bold.

| Category | Model | No. params (billions) | Humor & Sarcasm | Offensiveness | Sentiment & Emotion | Social Factors | Trustworthiness | CLS | PAIR | REG | SPAN | *Avg.* |
|---|---|---|---|---|---|---|---|---|---|---|---|---|
| baseline | majority | | 0.27 | 0.42 | 0.12 | 0.25 | 0.41 | 0.39 | 0.34 | 0.50 | 0.00 | 0.32 |
| | random | | 0.40 | 0.35 | 0.17 | 0.36 | 0.35 | 0.38 | 0.51 | 0.50 | 0.00 | 0.32 |
| zero-shot | EleutherAI-gpt-j-6b | 6 | 0.39 | 0.35 | 0.29 | 0.33 | 0.28 | 0.32 | 0.26 | 0.50 | 0.08 | 0.32 |
| | alpaca-native | 7 | 0.39 | 0.44 | 0.45 | 0.55 | 0.31 | 0.42 | 0.48 | 0.57 | 0.17 | 0.43 |
| | bigscience-bloom-560m | 0.56 | 0.30 | 0.24 | 0.14 | 0.26 | 0.18 | 0.21 | 0.26 | 0.49 | 0.07 | 0.22 |
| | bigscience-bloom-1b1 | 1 | 0.26 | 0.39 | 0.17 | 0.25 | 0.33 | 0.33 | 0.21 | 0.48 | 0.11 | 0.31 |
| | bigscience-bloom-3b | 3 | 0.37 | 0.44 | 0.30 | 0.34 | 0.30 | 0.37 | 0.32 | 0.51 | 0.12 | 0.37 |
| | bigscience-bloom-7b1 | 7 | 0.42 | 0.27 | 0.30 | 0.29 | 0.20 | 0.25 | 0.39 | 0.51 | 0.11 | 0.28 |
| | bigscience-bloomz-560m | 0.56 | 0.38 | 0.42 | 0.34 | 0.42 | 0.41 | 0.40 | 0.41 | 0.51 | 0.10 | 0.40 |
| | bigscience-bloomz-1b1 | 1 | 0.41 | 0.44 | 0.38 | 0.43 | 0.41 | 0.42 | 0.44 | 0.52 | 0.10 | 0.42 |
| | bigscience-bloomz-7b1 | 7 | 0.50 | 0.49 | 0.43 | 0.53 | 0.45 | 0.49 | 0.51 | 0.56 | 0.09 | 0.48 |
| | cerebras-Cerebras-GPT-111M | 0.11 | 0.21 | 0.07 | 0.18 | 0.09 | 0.03 | 0.05 | 0.18 | 0.49 | 0.07 | 0.10 |
| | cerebras-Cerebras-GPT-590M | 0.59 | 0.31 | 0.11 | 0.14 | 0.19 | 0.09 | 0.13 | 0.23 | 0.00 | 0.08 | 0.14 |
| | cerebras-Cerebras-GPT-2.7B | 2.7 | 0.32 | 0.19 | 0.14 | 0.23 | 0.16 | 0.17 | 0.26 | 0.49 | 0.18 | 0.19 |
| | cerebras-Cerebras-GPT-6.7B | 6.7 | 0.42 | 0.39 | 0.30 | 0.36 | 0.34 | 0.35 | 0.33 | 0.52 | 0.13 | 0.36 |
| | decapoda-research-llama-7b | 7 | 0.45 | 0.44 | 0.34 | 0.41 | 0.22 | 0.38 | 0.34 | 0.52 | 0.12 | 0.38 |
| | decapoda-research-llama-13b | 13 | 0.49 | 0.43 | 0.36 | 0.42 | 0.31 | 0.38 | 0.52 | 0.53 | 0.17 | 0.40 |
| | facebook-opt-1.3b | 1.3 | 0.41 | 0.43 | 0.28 | 0.33 | 0.39 | 0.38 | 0.34 | 0.50 | 0.12 | 0.38 |
| | facebook-opt-2.7b | 2.7 | 0.37 | 0.45 | 0.29 | 0.36 | 0.32 | 0.37 | 0.34 | 0.52 | 0.18 | 0.38 |
| | facebook-opt-6.7b | 6.7 | 0.20 | 0.31 | 0.20 | 0.16 | 0.17 | 0.21 | 0.13 | 0.48 | 0.16 | 0.24 |
| | facebook-opt-13b | 13 | 0.31 | 0.40 | 0.19 | 0.22 | 0.28 | 0.31 | 0.25 | 0.49 | 0.13 | 0.31 |
| | google-flan-t5-small | 0.08 | 0.40 | 0.43 | 0.30 | 0.33 | 0.41 | 0.39 | 0.43 | 0.49 | 0.06 | 0.39 |
| | google-flan-t5-base | 0.25 | 0.42 | 0.46 | 0.35 | 0.36 | 0.41 | 0.43 | 0.42 | 0.52 | 0.05 | 0.42 |
| | google-flan-t5-large | 0.78 | 0.51 | 0.52 | 0.47 | 0.50 | 0.47 | 0.51 | 0.58 | 0.54 | 0.09 | 0.50 |
| | google-flan-t5-xl | 3 | 0.57 | 0.53 | 0.48 | 0.55 | 0.47 | 0.53 | 0.51 | 0.60 | 0.14 | 0.51 |
| | google-flan-t5-xxl | 11 | 0.61 | 0.55 | 0.52 | 0.61 | 0.47 | 0.55 | 0.59 | 0.62 | 0.18 | 0.54 |
| | mosaicml-mpt-7b | 7 | 0.41 | 0.42 | 0.36 | 0.44 | 0.32 | 0.37 | 0.45 | 0.54 | 0.25 | 0.39 |
| | mosaicml-mpt-7b-instruct | 7 | 0.20 | 0.25 | 0.30 | 0.34 | 0.15 | 0.22 | 0.25 | 0.46 | 0.24 | 0.25 |
| | t5-small | 0.06 | 0.33 | 0.08 | 0.08 | 0.25 | 0.07 | 0.12 | 0.09 | 0.00 | 0.03 | 0.12 |
| | t5-base | 0.22 | 0.41 | 0.15 | 0.14 | 0.28 | 0.12 | 0.17 | 0.33 | 0.00 | 0.03 | 0.17 |
| | t5-large | 0.77 | 0.37 | 0.13 | 0.14 | 0.25 | 0.12 | 0.14 | 0.32 | 0.48 | 0.03 | 0.16 |
| | t5-3b | 3 | 0.13 | 0.21 | 0.14 | 0.17 | 0.23 | 0.19 | 0.19 | 0.48 | 0.09 | 0.19 |
| | t5-11b | 11 | 0.38 | 0.38 | 0.14 | 0.31 | 0.36 | 0.33 | 0.23 | 0.50 | 0.03 | 0.32 |
| | togethercomputer-RedPajama-INCITE-7B-Instruct | 7 | 0.41 | 0.48 | 0.37 | 0.41 | 0.42 | 0.44 | 0.39 | 0.53 | 0.11 | 0.43 |
| | llama2-7b-chat | 7 | 0.39 | 0.27 | 0.33 | 0.34 | 0.24 | 0.25 | 0.38 | 0.56 | 0.18 | 0.30 |
| | GPT-3.5 | | 0.64 | 0.56 | 0.57 | 0.65 | 0.45 | 0.57 | 0.49 | 0.67 | 0.21 | 0.56 |
| full | bert-base-uncased | 0.11 | 0.78 | 0.76 | 0.65 | 0.70 | 0.62 | 0.70 | 0.79 | 0.77 | 0.55 | 0.71 |
| | roberta-base | 0.086 | 0.79 | 0.77 | 0.68 | 0.72 | 0.63 | 0.70 | 0.83 | 0.79 | **0.64** | 0.72 |
| | deberta-v3 | 0.098 | **0.83** | **0.77** | **0.70** | **0.72** | **0.66** | **0.72** | **0.87** | **0.79** | 0.63 | **0.73** |
| | MiniLM | 0.066 | 0.77 | 0.72 | 0.61 | 0.67 | 0.58 | 0.66 | 0.78 | 0.69 | 0.57 | 0.67 |
| | T5* | 0.25 | 0.53 | 0.71 | 0.48 | 0.50 | 0.47 | 0.63 | 0.44 | 0.37 | 0.54 | 0.58 |

Table 5: A comparison of the benchmark performances of different models and training schemes on the SOCKETTE test set (a subset of SOCKET). Best-performing instances are shown in bold.

| Model | Group | No. params (billions) | Humor & Sarcasm | Offensiveness | Sentiment & Emotion | Social Factors | Trustworthiness | CLS | PAIR | REG | SPAN | *Total ratio* |
|---|---|---|---|---|---|---|---|---|---|---|---|---|
| chavinlo-alpaca-native | | 13.00 | 1.00 | 1.00 | 1.00 | 1.00 | 0.99 | 1.00 | 1.00 | 1.00 | 0.96 | 1.00 |
| bigscience-bloom-560m | | 0.56 | 0.49 | 0.74 | 0.42 | 0.54 | 0.76 | 0.71 | 0.72 | 0.00 | 0.80 | 0.63 |
| bigscience-bloom-1b1 | | 1 | 0.54 | 0.83 | 0.53 | 0.64 | 0.94 | 0.86 | 0.56 | 0.02 | 0.91 | 0.74 |
| bigscience-bloom-3b | | 3 | 0.89 | 0.96 | 0.75 | 0.89 | 0.97 | 0.99 | 0.92 | 0.29 | 0.95 | 0.90 |
| bigscience-bloom-7b1 | | 7 | 0.94 | 0.97 | 0.78 | 0.76 | 0.99 | 0.96 | 0.99 | 0.52 | 0.96 | 0.91 |
| bigscience-bloomz-560m | | 0.56 | 1.00 | 0.99 | 0.96 | 1.00 | 0.99 | 0.99 | 1.00 | 1.00 | 0.92 | 0.99 |
| bigscience-bloomz-1b1 | | 1 | 1.00 | 0.99 | 0.96 | 1.00 | 0.99 | 0.99 | 1.00 | 0.99 | 0.89 | 0.98 |
| bigscience-bloomz-7b1 | | 7 | 1.00 | 1.00 | 0.99 | 1.00 | 0.99 | 1.00 | 1.00 | 1.00 | 0.97 | 0.99 |
| google-flan-t5-small | | 0.08 | 0.94 | 0.96 | 0.90 | 0.94 | 0.97 | 0.95 | 0.92 | 1.00 | 0.79 | 0.94 |
| google-flan-t5-base | | 0.25 | 1.00 | 0.98 | 0.90 | 1.00 | 0.90 | 0.98 | 0.75 | 0.98 | 0.79 | 0.95 |
| google-flan-t5-large | | 0.78 | 1.00 | 1.00 | 0.91 | 1.00 | 0.90 | 0.97 | 0.75 | 1.00 | 0.98 | 0.96 |
| google-flan-t5-xl | | 3 | 1.00 | 1.00 | 0.92 | 1.00 | 0.93 | 0.98 | 0.84 | 1.00 | 0.95 | 0.97 |
| google-flan-t5-xxl | | 11 | 1.00 | 1.00 | 0.92 | 1.00 | 1.00 | 0.98 | 1.00 | 1.00 | 0.99 | 0.98 |
| cerebras-Cerebras-GPT-111M | | 0.11 | 0.30 | 0.22 | 0.15 | 0.16 | 0.27 | 0.20 | 0.41 | 0.01 | 0.66 | 0.21 |
| cerebras-Cerebras-GPT-590M | | 0.59 | 0.66 | 0.70 | 0.51 | 0.64 | 0.68 | 0.70 | 0.66 | 0.32 | 0.54 | 0.64 |
| cerebras-Cerebras-GPT-2.7B | | 2.70 | 0.73 | 0.88 | 0.66 | 0.62 | 0.88 | 0.77 | 0.60 | 0.98 | 0.96 | 0.79 |
| cerebras-Cerebras-GPT-6.7B | | 6.70 | 0.83 | 0.91 | 0.58 | 0.72 | 0.99 | 0.89 | 0.81 | 0.36 | 0.96 | 0.82 |
| EleutherAI-gpt-j-6b | | 6 | 0.70 | 0.77 | 0.56 | 0.65 | 0.74 | 0.75 | 0.53 | 0.41 | 0.88 | 0.70 |
| decapoda-research-llama-7b-hf | | 7 | 0.73 | 0.93 | 0.51 | 0.78 | 0.87 | 0.82 | 0.99 | 0.50 | 0.87 | 0.80 |
| decapoda-research-llama-13b-hf | | 13 | 0.44 | 0.48 | 0.62 | 0.50 | 0.57 | 0.46 | 0.57 | 0.75 | 0.95 | 0.53 |
| mosaicml-mpt-7b | | 7 | 0.87 | 0.99 | 0.67 | 0.96 | 1.00 | 0.95 | 0.86 | 0.63 | 0.96 | 0.91 |
| mosaicml-mpt-7b-instruct | | 7 | 0.36 | 0.62 | 0.72 | 0.75 | 0.67 | 0.66 | 0.53 | 0.46 | 0.99 | 0.64 |
| facebook-opt-1.3b | | 1.30 | 0.80 | 0.95 | 0.61 | 0.79 | 0.97 | 0.90 | 0.88 | 0.48 | 0.87 | 0.85 |
| facebook-opt-2.7b | | 2.70 | 0.82 | 0.94 | 0.65 | 0.83 | 1.00 | 0.98 | 1.00 | 0.05 | 0.87 | 0.87 |
| facebook-opt-6.7b | | 6.70 | 0.42 | 0.58 | 0.24 | 0.18 | 0.45 | 0.41 | 0.34 | 0.34 | 0.90 | 0.43 |
| facebook-opt-13b | | 13 | 0.65 | 0.82 | 0.30 | 0.50 | 0.73 | 0.70 | 0.68 | 0.20 | 0.74 | 0.64 |
| togethercomputer-RedPajama-INCITE-7B-Instruct | | 7 | 1.00 | 1.00 | 1.00 | 1.00 | 0.99 | 1.00 | 1.00 | 1.00 | 0.95 | 1.00 |
| t5-small | | 0.06 | 0.77 | 0.76 | 0.16 | 0.66 | 0.60 | 0.74 | 0.28 | 0.00 | 0.21 | 0.59 |
| t5-base | | 0.22 | 0.80 | 0.88 | 0.23 | 0.67 | 0.84 | 0.84 | 0.75 | 0.00 | 0.19 | 0.70 |
| t5-large | | 0.77 | 0.95 | 0.88 | 0.24 | 0.67 | 0.85 | 0.84 | 0.72 | 0.14 | 0.23 | 0.71 |
| t5-3b | | 3 | 0.40 | 0.39 | 0.46 | 0.50 | 0.54 | 0.46 | 0.49 | 0.23 | 0.58 | 0.44 |
| t5-11b | | 11 | 0.80 | 0.74 | 0.23 | 0.61 | 0.87 | 0.75 | 0.56 | 0.04 | 0.58 | 0.64 |
| llama2-7b-chat | | 7 | 1.00 | 0.99 | 0.99 | 1.00 | 1.00 | 0.99 | 0.99 | 1.00 | 0.99 | 0.99 |
| gpt-3.5 | | | 1.00 | 1.00 | 0.99 | 1.00 | 0.92 | 1.00 | 0.80 | 0.99 | 1.00 | 0.98 |
| Overall | | | 0.53 | 0.57 | 0.39 | 0.44 | 0.57 | 0.51 | 0.4 | 0.39 | 1.0 | 0.51 |

Table 6: The fraction of samples that each LLM can make inferences given the instruction prompts, when tested in a zero-shot setting.

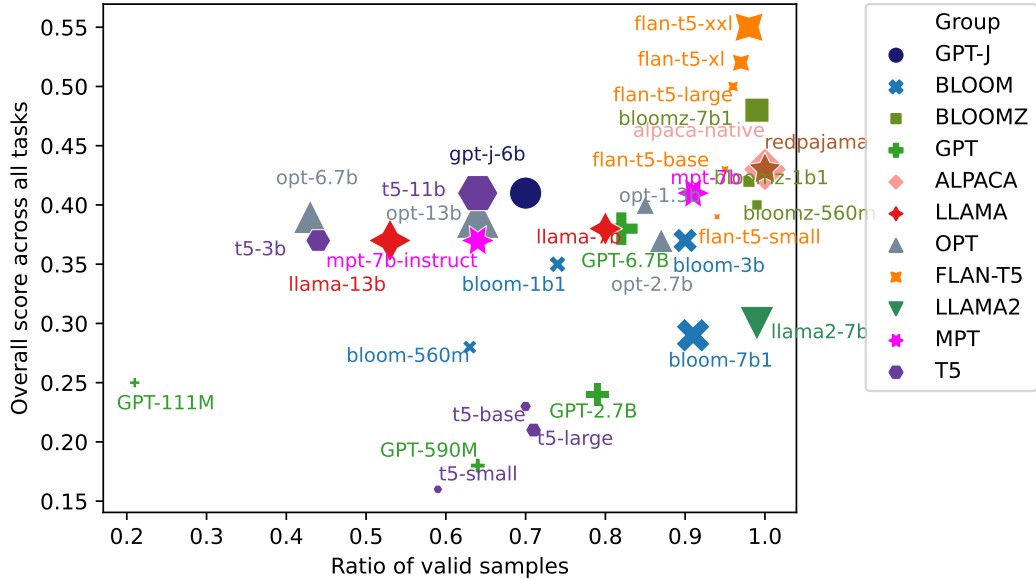

Figure 3: A comparison of the ratio of valid samples which the LLM was able to make an inference given the correct instruction prompt (x-axis) versus the overall scores when limited to the samples that the model was capable of making an inference (y-axis).

made after each training epoch on the validation set using Pearson's r correlation added by 1 and divided by 2 for regression tasks and macro F1 score for all other tasks. If there were multiple tasks considered due to multi-task training, the average of all task performances was used as the final validation score.

## B.5 Details on prompt-based finetuning (§4, §5)

We use fix prompts fine-tuning for all the prompt-based models. The batch size was set as 32 for training. For every single task, we set 10 as the max epoch and do early stopping based on the validation loss. The learning rate is set as 5e-5.

For classification tasks, the model is fine-tuned to generate the target label. For regression tasks, we first normalized the scores into (0,1) and then split the labels into two groups. The model is fine-

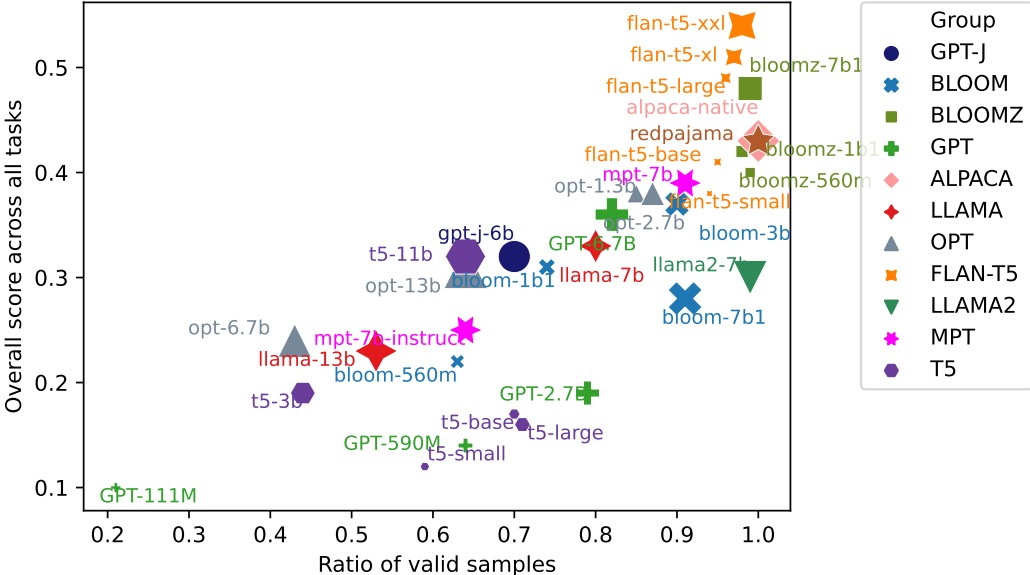

Figure 4: A comparison of the ratio of valid samples which the LLM was able to make an inference given the correct instruction prompt (x-axis) versus the overall scores across every sample in the test dataset where failed predictions are considered incorrect (y-axis).

| task | SOCKET | SOCKETTE |
|------|--------|----------|
| contextual-abuse#IdentityDirectedAbuse | 0.62 | 0.58 |
| contextual-abuse#PersonDirectedAbuse | 0.53 | 0.49 |
| crowdflower | 0.24 | 0.22 |
| dailydialog | 0.38 | 0.38 |
| hasbiasedimplication | 0.86 | 0.87 |
| hateoffensive | 0.93 | 0.93 |
| humor-pairs | 0.98 | 0.97 |
| implicit-hate#explicit_hate | 0.72 | 0.68 |
| implicit-hate#implicit_hate | 0.71 | 0.72 |
| implicit-hate#incitement_hate | 0.68 | 0.70 |
| implicit-hate#inferiority_hate | 0.60 | 0.69 |
| implicit-hate#stereotypical_hate | 0.68 | 0.68 |
| implicit-hate#threatening_hate | 0.63 | 0.67 |
| implicit-hate#white_grievance_hate | 0.70 | 0.68 |
| intentyn | 0.75 | 0.73 |
| jigsaw#identity_hate | 0.80 | 0.83 |
| jigsaw#insult | 0.88 | 0.87 |
| jigsaw#obscene | 0.91 | 0.91 |
| jigsaw#severe_toxic | 0.73 | 0.74 |
| jigsaw#threat | 0.85 | 1.00 |
| jigsaw#toxic | 0.90 | 0.90 |
| neutralizing-bias-pairs | 0.98 | 0.98 |
| offensiveyn | 0.82 | 0.82 |
| sarc | 0.74 | 0.77 |
| sentitreebank | 0.97 | 0.97 |
| sexyn | 0.79 | 0.77 |
| toxic-span | 0.68 | 0.69 |
| tweet_emoji | 0.34 | 0.32 |
| tweet_emotion | 0.81 | 0.81 |
| tweet_sentiment | 0.71 | 0.69 |
| two-to-lie#receiver_truth | 0.59 | 0.60 |
| two-to-lie#sender_truth | 0.58 | 0.58 |

Table 7: A comparison of the evaluation scores between the test sets for SOCKET versus SOCKETTE when evaluated on a DeBERTa-v3 model trained on a single-task setting.

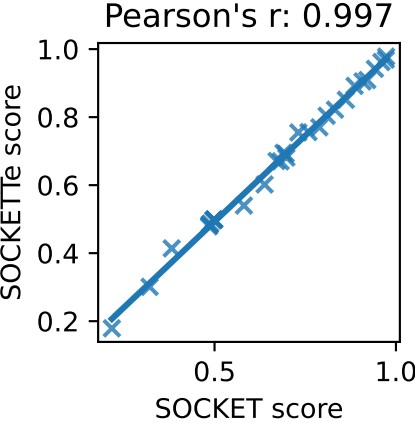

Figure 5: For each of the 32 tasks in SOCKET containing more than 1,000 test samples, we evaluate the performance of a deberta-v3 model trained on a single SOCKET task on both the original test set as well as the smaller SOCKETTE variant. The correlation between the two scores results in a high Pearson's r score of 0.997, indicating SOCKETTE can be reliably deployed for more rapid model testing.

tuned to predict "yes" or "no" regarding the prompt question. During inference, the probability of the "yes" token is used as the prediction score. For span tasks, we directly train the model to generate the full answer.

## B.6 Details on zero-shot predictions (§4, §5)

We use manually designed prompts for all the zero-shot prediction tasks and the prompts are shown in Table 8.

## B.7 Computing correlation scores of task dependencies (§5)

Because our framework consists of several task types, it is challenging to obtain a unified metric of correlation across different task comparisons. We use the following rules to obtain correlation values:

- Regression task & regression task: We compute the Pearson's correlation coefficient of the two arrays.

- Regression task & binary classification task: We compute the point biserial correlation coefficient of a continuous array and a binary array.

- Regression task & multi-class classification task: We set up a linear regression task using the one-hot coded values of the multi-class array as independent variables and the continuous array as the dependent variable. We report the root of the R-squared value of the regression as correlation (Olsson et al., 1982).

- Binary classification task & binary classification task: We compute the Matthews' correlation coefficient (Matthews, 1975) from the two binary arrays.

- Binary or multi-task classification task & multi-class classification task: We compute the Cramer's V score (Cramér, 1999) from the two arrays of categorical variables.

## B.8 Computing pairwise model similarities (§5)

We quantify the model similarity between two tasks as follows. We finetune a pretrained LLM on task $t_A$ to obtain a model $m_A$, and another LLM on task $t_B$ to obtain $m_B$. We obtain pairwise model similarities by inferring both models on a sufficiently large dataset—in this case the entire test set of all tasks—and computing the correlation of the two inferred arrays. We construct an undirected graph (Figure 7) where the thickness and color represent absolute correlation strength and polarity between the two models. The addition of polarity enables us to further discover strong negative correlations with task pairs such as politeness and offensiveness.

## B.9 List of all potential tasks and datasets for SOCKET

| Type | Task | Question/Options | |
|------|------|------------------|---|
| PAIR | talkdown-pairs | For the quote "text_a" and its context "text_b", is the quote condescending? | ['No', 'Yes'] |
| REG | hahackathon#humor_rating | Determine the degree of humor of the given sentence: "text". The score should be ranging from 0.0 to 5.0, and can be a decimal. 0 is not humorous at all, and 5 is very humorous. | |
| CLS | hahackathon#is_humor | For the sentence: "text", is it humorous? | ['No', 'Yes'] |
| PAIR | humor-pairs | The first sentence is "text_a". The second sentence is "text_b". Is the first sentence funnier than the second sentence? | ['Yes', 'No'] |
| CLS | sarc | For the sentence: "text", is it sarcastic? | ['Yes', 'No'] |
| CLS | tweet_irony | For the sentence: "text", is it ironic? | ['No', 'Yes'] |
| CLS | contextual-abuse#IdentityDirectedAbuse | For the sentence: "text", is it identity directed abuse? | ['No', 'Yes'] |
| CLS | contextual-abuse#PersonDirectedAbuse | For the sentence: "text", is it person directed abuse? | ['No', 'Yes'] |
| REG | hahackathon#offense_rating | Determine the degree of offense of the given sentence: "text". The score should be ranging from 0.0 to 5.0, and can be a decimal. | |
| CLS | hasbiasedimplication | For the sentence: "text", does it imply some biases? | ['No', 'Yes'] |
| CLS | hateoffensive | For the sentence: "text", is it hate or offensive? | ['Hate', 'Offensive', 'Neither'] |
| CLS | implicit-hate#explicit_hate | For the sentence: "text", is it explicit hate? | ['No', 'Yes'] |
| CLS | implicit-hate#implicit_hate | For the sentence: "text", is it implicitly hateful? | ['No', 'Yes'] |
| CLS | implicit-hate#incitement_hate | For the sentence: "text", is it a hateful incitement to act? | ['No', 'Yes'] |
| CLS | implicit-hate#inferiority_hate | For the sentence: "text", is it inferiority hate? | ['No', 'Yes'] |
| CLS | implicit-hate#stereotypical_hate | For the sentence: "text", is it a hateful message involving stereotypes? | ['No', 'Yes'] |
| CLS | implicit-hate#threatening_hate | For the sentence: "text", is it hateful in a threatening way? | ['No', 'Yes'] |
| CLS | implicit-hate#white_grievance_hate | For the sentence: "text", is it white grievance hate? | ['No', 'Yes'] |
| CLS | intentyn | For the sentence: "text", is it intentional? | ['No', 'Yes'] |
| CLS | jigsaw#identity_hate | For the sentence: "text", is it identity hate? | ['No', 'Yes'] |
| CLS | jigsaw#insult | For the sentence: "text", is it an insult? | ['No', 'Yes'] |
| CLS | jigsaw#obscene | For the sentence: "text", is it obscene? | ['No', 'Yes'] |
| CLS | jigsaw#severe_toxic | For the sentence: "text", is it severely toxic? | ['No', 'Yes'] |
| CLS | jigsaw#threat | For the sentence: "text", is it a threat? | ['No', 'Yes'] |
| CLS | jigsaw#toxic | For the sentence: "text", is it toxic? | ['No', 'Yes'] |
| CLS | offensiveyn | For the sentence: "text", is it offensive? | ['No', 'Yes'] |
| CLS | sexyn | For the sentence: "text", is it sexist? | ['No', 'Yes'] |
| SPAN | toxic-span | In the sentence: "text", which part of it can be identified as toxic? | |
| CLS | tweet_offensive | For the sentence: "text", is it offensive? | ['No', 'Yes'] |
| CLS | crowdflower | For the sentence: "text", what is its emotion? | ['empty', 'sadness', 'enthusiasm', 'neutral', 'worry', 'love', 'fun', 'hate', 'happiness', 'relief', 'boredom', 'surprise', 'anger'] |
| CLS | dailydialog | For the given conversation, "text", what is its emotion? | ['no emotion', 'anger', 'disgust', 'fear', 'happiness', 'sadness', 'surprise'] |
| REG | emobank#arousal | Given the VAD model of emotion, determine the degree of arousal of the given sentence: "text". The score should be ranging from 0.0 to 5.0, and can be a decimal. | |
| REG | emobank#dominance | Given the VAD model of emotion, determine the degree of dominance of the given sentence: "text". The score should be ranging from 0.0 to 5.0, and can be a decimal. | |
| REG | emobank#valence | Given the VAD model of emotion, determine the degree of valence of the given sentence: "text". The score should be ranging from 0.0 to 5.0, and can be a decimal. | |
| SPAN | emotion-span | In the sentence: "text", which part of it expresses strong emotion? | |
| REG | empathy#distress | Determine the degree of distress of the given sentence: "text". The score should be ranging from 0.0 to 7.0, and can be a decimal. | |
| CLS | empathy#distress_bin | For the sentence: "text", is it showing distress? | ['No', 'Yes'] |
| PAIR | same-side-pairs | For the sentences: "text_a" and "text_b", are they on the same side? | ['No', 'Yes'] |
| CLS | sentitreebank | For the sentence: "text", is it positive? | ['Yes', 'No'] |
| CLS | tweet_emoji | For the sentence: "text", what is the emoji that can be added to it? | 20 emojis |
| CLS | tweet_emotion | For the sentence: "text", what is its emotion? | ['anger', 'joy', 'optimism', 'sadness'] |
| CLS | tweet_sentiment | For the sentence: "text", what is its sentiment? | ['negative', 'neutral', 'positive'] |
| CLS | complaints | For the sentence: "text", is it a complaint? | ['No', 'Yes'] |
| REG | empathy#empathy | Determine the degree of empathy of the given sentence: "text". The score should be ranging from 0.0 to 7.0, and can be a decimal. | |
| CLS | empathy#empathy_bin | For the sentence: "text", is it expressing empathy? | ['No', 'Yes'] |
| CLS | hayati_politeness | For the sentence: "text", is it polite? | ['No', 'Yes'] |
| CLS | questionintimacy | For the sentence: "text", how intimate do you think it is? | ['Very intimate', 'Intimate', 'Somewhat intimate', 'Not very intimate', 'Not intimate', 'Not intimate at all'] |
| CLS | stanfordpoliteness | For the sentence: "text", is it polite? | ['Yes', 'No'] |
| CLS | bragging#brag_achievement | For the sentence: "text", is it bragging about an achievement? | ['No', 'Yes'] |
| CLS | bragging#brag_action | For the sentence: "text", is it bragging about an action? | ['No', 'Yes'] |
| CLS | bragging#brag_possession | For the sentence: "text", is it bragging about a possession? | ['No', 'Yes'] |
| CLS | bragging#brag_trait | For the sentence: "text", is it bragging about a trait? | ['No', 'Yes'] |
| CLS | hypo-l | For the sentence: "text", is it a hyperbole? | ['No', 'Yes'] |
| PAIR | neutralizing-bias-pairs | For the sentences: "text_a" and "text_b", which one is biased? | ['the first sentence is biased', 'the second sentence is biased'] |
| SPAN | propaganda-span | In the sentence: "text", which part of it can be identified as the propaganda? | |
| CLS | rumor#rumor_bool | For the sentence: "text", is it a rumor? | ['No', 'Yes'] |
| CLS | two-to-lie#receiver_truth | For the sentence :"text", will it be perceived as a lie by the receiver? | ['Yes', 'No'] |
| CLS | two-to-lie#sender_truth | For the sentence :"text", is the sender intending to tell a lie? | ['Yes', 'No'] |

Table 8: The manually designed prompt questions and options used for each task.

| Type | Task | Question |
|------|------|----------|
| REG | hahackathon#humor_rating | For the sentence:"text", is it humorous? |
| REG | hahackathon#offense_rating | For the sentence:"text", is it offensive? |
| REG | emobank#arousal | For the sentence:"text", is the presented emotion highly arousal? |
| REG | emobank#dominance | For the sentence:"text", is the presented emotion highly dominant? |
| REG | emobank#valence | For the sentence:"text", is the presented emotion positive? |
| REG | empathy#distress | For the sentence:"text", does it show distress? |
| REG | empathy#empathy | For the sentence:"text", does it show empathy? |

Table 9: The manually designed prompt questions for fine-tuning regression tasks over the t5 model.

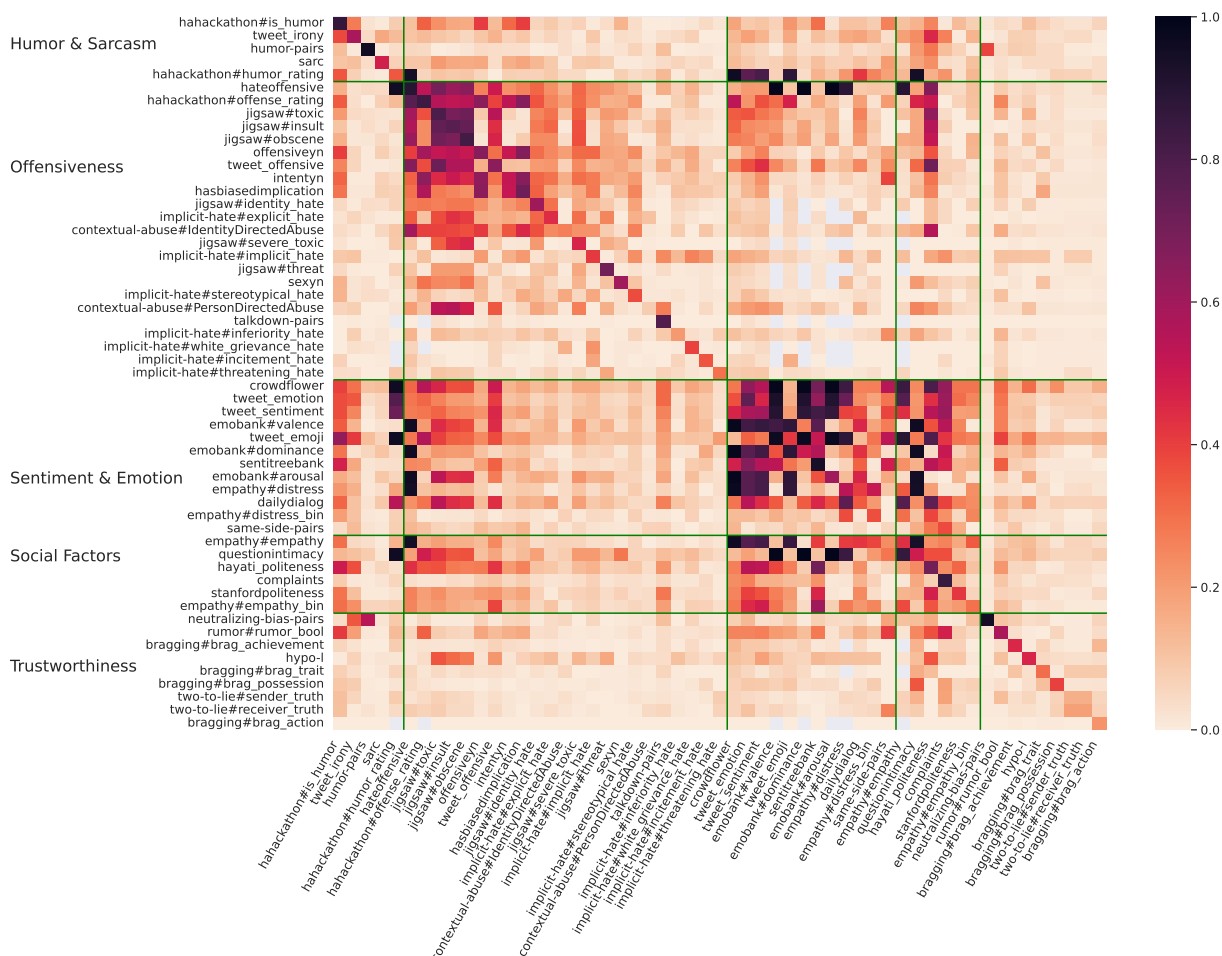

Figure 6: A detailed heatmap of Figure 2 showing task dependency among all task pairs as well as task labels. Each value represents the absolute strength of correlation between the true labels of the test set of a specific task (columns) and the predictions made on that task using a model trained on a different task (rows).

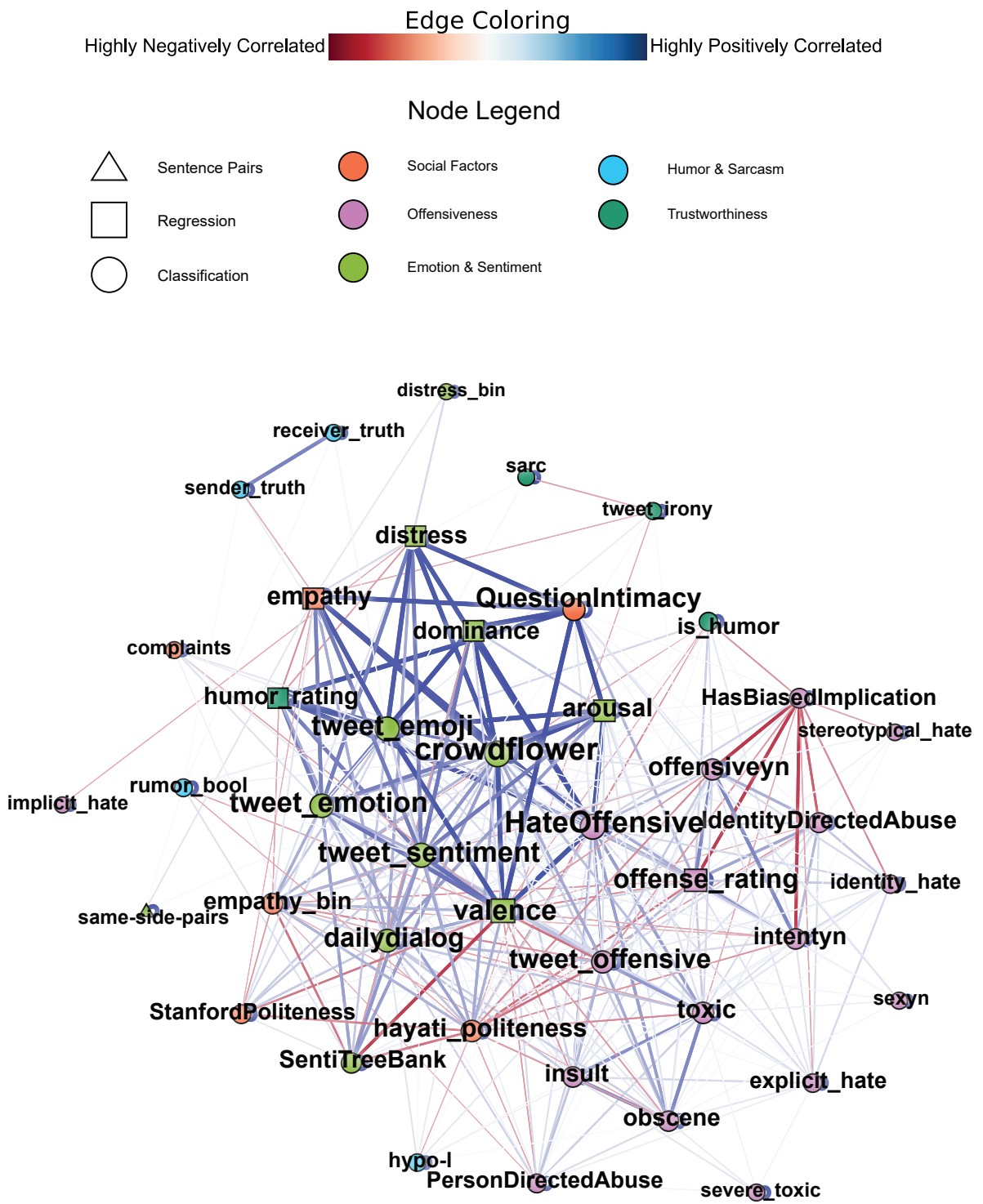

Figure 7: Weighted, undirected graph of model correlations. Each edge between nodes $i$ and $j$ is weighted by the correlation between predictions from a model fine-tuned on task $i$ and predictions from a model fine-tuned on task $j$, evaluated on the entire SOCKET dataset. Nodes are sized proportionally to their weighted degree and a Yifan Hu algorithm is applied for layout, with minor adjustments for readability. Refer to §B.8 for details on how the pairwise score for each edge was computed. We observe strong positive correlations similar to Figure 2, especially within the Sentiment & Emotion category and the Offensiveness category. We also see that across-category transfers may happen in a negative direction such as *hayati_politeness* and several Offensiveness tasks.

| Dataset | Task name | Task type | Rand. | Maj. | Single task | Categorywise | All tasks |
|---|---|---|---|---|---|---|---|
| *Humor & Sarcasm* | | | | | | | |
| hahackathon | humor_rating | REG | 0.5 | 0.5 | 0.68 0.01 | 0.67 0.01 | 0.66 0.03 |
| hahackathon | is_humor | CLS | 0.49 | 0.38 | 0.93 0.00 | 0.93 0.00 | 0.91 0.01 |
| humor-pairs | humor-pairs | PAIR | 0.50 | 0.34 | 0.98 0.00 | 0.98 0.00 | 0.97 0.00 |
| tweet_irony | tweet_irony | CLS | 0.49 | 0.28 | 0.80 0.01 | 0.80 0.00 | 0.75 0.03 |
| *Offensiveness* | | | | | | | |
| contextual-abuse | IdentityDirectedAbuse | CLS | 0.35 | 0.50 | 0.61 0.02 | **0.62** 0.00 | **0.63** 0.01 |
| contextual-abuse | PersonDirectedAbuse | CLS | 0.34 | 0.50 | 0.54 0.01 | **0.55** 0.02 | **0.57** 0.02 |
| hahackathon | offense_rating | REG | 0.5 | 0.5 | 0.91 0.01 | 0.91 0.00 | **0.92** 0.01 |
| hasbiasedimplication | hasbiasedimplication | CLS | 0.50 | 0.37 | 0.86 0.00 | 0.86 0.00 | **0.87** 0.00 |
| hateoffensive | hateoffensive | CLS | 0.27 | 0.27 | 0.93 0.00 | **0.95** 0.01 | **0.95** 0.01 |
| implicit-hate | explicit_hate | CLS | 0.37 | 0.49 | 0.72 0.00 | **0.74** 0.01 | 0.73 0.01 |
| implicit-hate | implicit_hate | CLS | 0.47 | 0.43 | 0.71 0.01 | 0.71 0.01 | 0.69 0.01 |
| implicit-hate | incitement_hate | CLS | 0.38 | 0.49 | 0.67 0.01 | **0.68** 0.01 | 0.67 0.00 |
| implicit-hate | inferiority_hate | CLS | 0.34 | 0.50 | 0.59 0.02 | 0.59 0.01 | 0.58 0.02 |
| implicit-hate | stereotypical_hate | CLS | 0.37 | 0.49 | 0.68 0.01 | 0.66 0.01 | 0.68 0.01 |
| implicit-hate | threatening_hate | CLS | 0.33 | 0.50 | 0.60 0.02 | **0.64** 0.01 | **0.65** 0.04 |
| implicit-hate | white_grievance_hate | CLS | 0.39 | 0.48 | 0.71 0.01 | 0.71 0.01 | 0.71 0.02 |
| intentyn | intentyn | CLS | 0.47 | 0.42 | 0.75 0.00 | 0.74 0.01 | 0.74 0.01 |
| jigsaw | identity_hate | CLS | 0.34 | 0.50 | 0.79 0.01 | 0.76 0.01 | 0.79 0.01 |
| jigsaw | insult | CLS | 0.38 | 0.49 | 0.88 0.00 | 0.87 0.01 | 0.88 0.01 |
| jigsaw | obscene | CLS | 0.38 | 0.49 | 0.90 0.01 | **0.91** 0.00 | **0.91** 0.01 |
| jigsaw | severe_toxic | CLS | 0.34 | 0.50 | 0.74 0.01 | 0.73 0.02 | 0.71 0.02 |
| jigsaw | threat | CLS | 0.34 | 0.50 | 0.85 0.02 | 0.81 0.04 | 0.80 0.03 |
| jigsaw | toxic | CLS | 0.41 | 0.47 | 0.91 0.00 | 0.90 0.01 | 0.91 0.01 |
| offensiveyn | offensiveyn | CLS | 0.50 | 0.37 | 0.82 0.00 | **0.83** 0.01 | **0.83** 0.00 |
| sexyn | sexyn | CLS | 0.37 | 0.49 | 0.80 0.01 | 0.79 0.00 | 0.79 0.00 |
| talkdown-pairs | talkdown-pairs | PAIR | 0.50 | 0.33 | 0.89 0.01 | 0.88 0.01 | 0.88 0.00 |
| toxic-span | toxic-span | SPAN | 0 | 0 | 0.68 0.02 | 0.65 0.05 | 0.67 0.05 |
| tweet_offensive | tweet_offensive | CLS | 0.48 | 0.42 | 0.81 0.01 | 0.81 0.01 | 0.80 0.00 |
| *Sentiment & Emotion* | | | | | | | |
| crowdflower | crowdflower | CLS | 0.06 | 0.03 | 0.24 0.00 | 0.23 0.01 | 0.22 0.00 |
| dailydialog | dailydialog | CLS | 0.07 | 0.13 | 0.43 0.04 | **0.47** 0.00 | **0.47** 0.02 |
| emobank | arousal | REG | 0.5 | 0.5 | 0.80 0.02 | **0.81** 0.02 | 0.79 0.02 |
| emobank | dominance | REG | 0.5 | 0.5 | 0.75 0.02 | 0.75 0.01 | 0.73 0.02 |
| emobank | valence | REG | 0.5 | 0.5 | 0.92 0.01 | 0.92 0.01 | 0.90 0.01 |
| emotion-span | emotion-span | SPAN | 0 | 0 | 0.96 0.01 | 0.89 0.00 | 0.86 0.03 |
| empathy | distress | REG | 0.5 | 0.5 | 0.77 0.01 | 0.75 0.03 | 0.72 0.01 |
| empathy | distress_bin | CLS | 0.50 | 0.31 | 0.68 0.01 | **0.69** 0.03 | 0.65 0.02 |
| same-side-pairs | same-side-pairs | PAIR | 0.48 | 0.35 | 0.66 0.12 | **0.70** 0.09 | **0.76** 0.05 |
| sentitreebank | sentitreebank | CLS | 0.50 | 0.31 | 0.97 0.00 | 0.96 0.01 | 0.96 0.01 |
| tweet_emoji | tweet_emoji | CLS | 0.04 | 0.02 | 0.34 0.00 | 0.33 0.00 | 0.33 0.00 |
| tweet_emotion | tweet_emotion | CLS | 0.24 | 0.14 | 0.80 0.01 | **0.81** 0.01 | 0.80 0.00 |
| tweet_sentiment | tweet_sentiment | CLS | 0.32 | 0.22 | 0.71 0.00 | 0.71 0.01 | 0.69 0.01 |
| *Social Factors* | | | | | | | |
| complaints | complaints | CLS | 0.50 | 0.36 | 0.92 0.01 | 0.92 0.00 | 0.91 0.01 |
| empathy | empathy | REG | 0.5 | 0.5 | 0.70 0.04 | **0.71** 0.03 | 0.70 0.01 |
| empathy | empathy_bin | CLS | 0.50 | 0.33 | 0.63 0.02 | 0.62 0.01 | 0.59 0.03 |
| hayati_politeness | hayati_politeness | CLS | 0.47 | 0.41 | 0.87 0.01 | 0.87 0.05 | **0.89** 0.03 |
| questionintimacy | questionintimacy | CLS | 0.16 | 0.06 | 0.49 0.03 | 0.48 0.02 | 0.46 0.02 |
| stanfordpoliteness | stanfordpoliteness | CLS | 0.50 | 0.36 | 0.70 0.02 | **0.71** 0.01 | **0.72** 0.02 |
| *Trustworthiness* | | | | | | | |
| bragging | brag_achievement | CLS | 0.36 | 0.49 | 0.74 0.01 | 0.69 0.01 | **0.76** 0.03 |
| bragging | brag_action | CLS | 0.35 | 0.50 | 0.59 0.02 | 0.57 0.06 | 0.59 0.04 |
| bragging | brag_possession | CLS | 0.35 | 0.50 | 0.70 0.02 | 0.66 0.03 | 0.52 0.03 |
| bragging | brag_trait | CLS | 0.34 | 0.50 | 0.67 0.01 | 0.59 0.07 | 0.61 0.04 |
| hypo-l | hypo-l | CLS | 0.48 | 0.41 | 0.74 0.01 | 0.71 0.01 | 0.69 0.01 |
| neutralizing-bias-pairs | neutralizing-bias-pairs | PAIR | 0.50 | 0.33 | 0.96 0.01 | 0.96 0.01 | 0.96 0.00 |
| propaganda-span | propaganda-span | SPAN | 0 | 0 | 0.22 0.10 | **0.23** 0.03 | **0.24** 0.00 |
| rumor | rumor_bool | CLS | 0.49 | 0.39 | 0.85 0.05 | 0.78 0.02 | 0.78 0.05 |
| two-to-lie | receiver_truth | CLS | 0.38 | 0.49 | 0.57 0.02 | 0.57 0.02 | 0.53 0.01 |
| two-to-lie | sender_truth | CLS | 0.38 | 0.49 | 0.58 0.02 | **0.59** 0.03 | 0.55 0.03 |

Table 10: Detailed table of performance scores from comparing single-task vs multi-task trained models in Section 6 (refer to Table 3 in Section 6). There are no significant gains from the two multi-task settings in the Humor & Sarcasm category, where the tasks in general have low task dependency (ref. Section 5). However, for other categories we see several instances of tasks where multi-task trained model have greater performance.

| Paper/Dataset Title | Tasks | Reference |
| --- | --- | --- |
| Automatic Identification and Classification of Bragging in Social Media | Bragging (Achievement) | Jin et al. (2022) |
| Automatic Identification and Classification of Bragging in Social Media (Jin et al., 2022) | Bragging (Action) | Jin et al. (2022) |
| Automatic Identification and Classification of Bragging in Social Media | Bragging (Possession) | Jin et al. (2022) |
| Automatic Identification and Classification of Bragging in Social Media | Bragging (Trait) | Jin et al. (2022) |
| Automatically Identifying Complaints in Social Media | Complaints | Preoţiuc-Pietro et al. (2019) |
| Introducing CAD: the Contextual Abuse Dataset | Identity Based Hate | Vidgen et al. (2021) |
| Introducing CAD: the Contextual Abuse Dataset | Individual Hate | Vidgen et al. (2021) |
| Introducing CAD: the Contextual Abuse Dataset | Group-Based Hate | Vidgen et al. (2021) |
| Introducing CAD: the Contextual Abuse Dataset | Counter Speech | Vidgen et al. (2021) |
| Sentiment Analysis in Text | Emotion | CrowdFlower (2016) |
| DailyDialog: A Manually Labelled Multi-turn Dialogue Dataset | Emotion | Li et al. (2017) |
| EmoBank: Studying the Impact of Annotation Perspective and Representation Format on Dimensional Emotion Analysis | Emotion (Valence) | Buechel and Hahn (2017) |
| EmoBank: Studying the Impact of Annotation Perspective and Representation Format on Dimensional Emotion Analysis | Emotion (Arousal) | Buechel and Hahn (2017) |
| EmoBank: Studying the Impact of Annotation Perspective and Representation Format on Dimensional Emotion Analysis | Emotion (Dominance) | Buechel and Hahn (2017) |
| Detecting Emotion Stimuli in Emotion-Bearing Sentences | Emotion | Ghazi et al. (2015) |
| Measuring the Language of Self-Disclosure across Corpora | Disturbance | Reuel et al. (2022) |
| Measuring the Language of Self-Disclosure across Corpora | Empathy | Reuel et al. (2022) |
| SemEval 2021 Task 7: HaHackathon, Detecting and Rating Humor and Offense | Humor Rating | Meaney et al. (2021) |

| | | |
|---|---|---|
| SemEval 2021 Task 7: HaHackathon, Detecting and Rating Humor and Offense | Funny (boolean) | Meaney et al. (2021) |
| SemEval 2021 Task 7: HaHackathon, Detecting and Rating Humor and Offense | Offensiveness | Meaney et al. (2021) |
| Social Bias Frames: Reasoning about Social and Power Implications of Language | Biased Implication | Sap et al. (2020) |
| Social Bias Frames: Reasoning about Social and Power Implications of Language | Intent | Sap et al. (2020) |
| Social Bias Frames: Reasoning about Social and Power Implications of Language | Offensiveness | Sap et al. (2020) |
| Social Bias Frames: Reasoning about Social and Power Implications of Language | Sexism | Sap et al. (2020) |
| Automated Hate Speech Detection and the Problem of Offensive Language | Offensive | Davidson et al. (2017) |
| Does BERT Learn as Humans Perceive? Understanding Linguistic Styles through Lexica | Politeness | Hayati et al. (2021) |
| Does BERT Learn as Humans Perceive? Understanding Linguistic Styles through Lexica | Positivity | Hayati et al. (2021) |
| Does BERT Learn as Humans Perceive? Understanding Linguistic Styles through Lexica | Anger | Hayati et al. (2021) |
| Does BERT Learn as Humans Perceive? Understanding Linguistic Styles through Lexica | Disgust | Hayati et al. (2021) |
| Does BERT Learn as Humans Perceive? Understanding Linguistic Styles through Lexica | Fear | Hayati et al. (2021) |
| Does BERT Learn as Humans Perceive? Understanding Linguistic Styles through Lexica | Joy | Hayati et al. (2021) |
| Does BERT Learn as Humans Perceive? Understanding Linguistic Styles through Lexica | Sadness | Hayati et al. (2021) |
| SemEval-2020 Task 7: Assessing Humor in Edited News Headlines | Funnier Sequence | Hossain et al. (2020) |
| MOVER: Mask, Over-generate and Rank for Hyperbole Generation | Hyperbole | Zhang and Wan (2022) |
| Latent Hatred: A Benchmark for Understanding Implicit Hate Speech | Explicit Hate | ElSherief et al. (2021) |
| Latent Hatred: A Benchmark for Understanding Implicit Hate Speech | Implicit Hate | ElSherief et al. (2021) |
| Latent Hatred: A Benchmark for Understanding Implicit Hate Speech | Incitement | ElSherief et al. (2021) |
| Latent Hatred: A Benchmark for Understanding Implicit Hate Speech | Inferiority | ElSherief et al. (2021) |
| Latent Hatred: A Benchmark for Understanding Implicit Hate Speech | Stereotyping | ElSherief et al. (2021) |
| Latent Hatred: A Benchmark for Understanding Implicit Hate Speech | Threat | ElSherief et al. (2021) |
| Latent Hatred: A Benchmark for Understanding Implicit Hate Speech | Offensive | ElSherief et al. (2021) |
| Latent Hatred: A Benchmark for Understanding Implicit Hate Speech | Irony | ElSherief et al. (2021) |
| Latent Hatred: A Benchmark for Understanding Implicit Hate Speech | Other Hate | ElSherief et al. (2021) |

| | | |
|---|---|---|
| Toxic Comment Classification Challenge | Identity-Based Hate | Jigsaw (2017) |
| Toxic Comment Classification Challenge | Insult | Jigsaw (2017) |
| Toxic Comment Classification Challenge | Obscenity | Jigsaw (2017) |
| Toxic Comment Classification Challenge | Severe Toxicity | Jigsaw (2017) |
| Toxic Comment Classification Challenge | Threat | Jigsaw (2017) |
| Toxic Comment Classification Challenge | Toxicity | Jigsaw (2017) |
| Automatically Neutralizing Subjective Bias in Text | Bias | Pryzant et al. (2020) |
| SemEval-2020 Task 11: Detection of Propaganda Techniques in News Articles | Propaganda Technique | Da San Martino et al. (2020) |
| Quantifying Intimacy in Language | Intimacy | Pei and Jurgens (2020) |
| Detect Rumors in Microblog Posts Using Propagation Structure via Kernel Learning | Rumor Detection | Ma et al. (2017) |
| On Classifying whether Two Texts are on the Same Side of an Argument | Stance | Körner et al. (2021) |
| A Large Self-Annotated Corpus for Sarcasm | Sarcasm | Khodak et al. (2018) |
| Recursive Deep Models for Semantic Compositionality Over a Sentiment Treebank | Sentiment | Socher et al. (2013) |
| Facilitating the Communication of Politeness through Fine-Grained Paraphrasing | Politeness | Fu et al. (2020) |
| TalkDown: A Corpus for Condescension Detection in Context | Condescension | Wang and Potts (2019) |
| SemEval-2021 Task 5: Toxic Spans Detection | Toxicity | Pavlopoulos et al. (2021) |
| SemEval 2018 Task 2: Multilingual Emoji Prediction | Emoji | Barbieri et al. (2018) |
| SemEval-2018 Task 1: Affect in Tweets | Emotion | Mohammad et al. (2018) |
| SemEval-2018 Task 3: Irony Detection in English Tweets | Irony | Van Hee et al. (2018) |
| Predicting the Type and Target of Offensive Posts in Social Media | Offensiveness | Zampieri et al. (2019a) |
| SemEval-2017 Task 4: Sentiment Analysis in Twitter | Sentiment | Rosenthal et al. (2017) |
| It Takes Two to Lie: One to Lie, and One to Listen | Sender Truth | Peskov et al. (2020) |
| It Takes Two to Lie: One to Lie, and One to Listen | Receiver Truth | Peskov et al. (2020) |
| "So You Think You're Funny?": Rating the Humour Quotient in Standup Comedy | Humor Rating | Mittal et al. (2021) |
| DEBAGREEMENT: A comment-reply dataset for (dis)agreement detection in online debates | Stance | Pougué-Biyong et al. (2021) |
| The CLEF-2021 CheckThat! Lab on Detecting Check-Worthy Claims, Previously Fact-Checked Claims, and Fake News | Trustworthiness | Nakov et al. (2021) |
| Finding Deceptive Opinion Spam by Any Stretch of the Imagination | Deceipt | Ott et al. (2011) |
| Finding Deceptive Opinion Spam by Any Stretch of the Imagination | Fact | Ott et al. (2011) |
| A Clustering Approach for Nearly Unsupervised Recognition of Nonliteral Language | Nonliteral Language | Birke and Sarkar (2006) |
| Detecting Community Sensitive Norm Violations in Online Conversations | Community Norms | Park et al. (2021) |
| Can Machines Learn Morality? The Delphi Experiment | Moral Judgement | Jiang et al. (2021) |

| | | |
|---|---|---|
| SemEval-2019 Task 5: Multilingual Detection of Hate Speech Against Immigrants and Women in Twitter | Hate Speech | Basile et al. (2019) |
| SemEval-2019 Task 6: Identifying and Categorizing Offensive Language in Social Media (OffensEval) | Offensiveness | Zampieri et al. (2019b) |
| CivilComments | Toxicity | (Jigsaw, 2019) |
| CivilComments | Very Toxic | (Jigsaw, 2019) |
| (Male, Bachelor) and (Female, Ph.D) have different connotations: Parallelly Annotated Stylistic Language Dataset with Multiple Personas | Gender | Kang et al. (2019) |
| (Male, Bachelor) and (Female, Ph.D) have different connotations: Parallelly Annotated Stylistic Language Dataset with Multiple Personas | Age | Kang et al. (2019) |
| (Male, Bachelor) and (Female, Ph.D) have different connotations: Parallelly Annotated Stylistic Language Dataset with Multiple Personas | Country | Kang et al. (2019) |
| (Male, Bachelor) and (Female, Ph.D) have different connotations: Parallelly Annotated Stylistic Language Dataset with Multiple Personas | Political view | Kang et al. (2019) |
| (Male, Bachelor) and (Female, Ph.D) have different connotations: Parallelly Annotated Stylistic Language Dataset with Multiple Personas | Education | Kang et al. (2019) |
| (Male, Bachelor) and (Female, Ph.D) have different connotations: Parallelly Annotated Stylistic Language Dataset with Multiple Personas | Ethnicity | Kang et al. (2019) |
| Webis Clickbait Corbus 2017 | Clickbait | Potthast et al. (2018) |
| VU Amsterdam Metaphor Corpus | Metaphor | Steen et al. (2011) |
| Measuring Sentence-Level and Aspect-Level (Un)certainty in Science Communications | Uncertainty | Pei and Jurgens (2021) |
| Dear Sir or Madam, May I Introduce the GYAFC Dataset: Corpus, Benchmarks and Metrics for Formality Style Transfer | Formality | Rao and Tetreault (2018) |
| International Survey on Emotion Antecedents and Reactions | Sentiment | Scherer and Wallbott (1994) |
| Short Jokes | Joke | Moudgil |
| Short Text Corpus with Focus on Humor Detection | Joke | CrowdTruth (2016) |
| Hateful Symbols or Hateful People? Predictive Features for Hate Speech Detection on Twitter | Sexism | Waseem and Hovy (2016) |
| Hateful Symbols or Hateful People? Predictive Features for Hate Speech Detection on Twitter | Racism | Waseem and Hovy (2016) |
| Studying the Dark Triad of Personality through Twitter Behavior | narcissism | Preoţiuc-Pietro et al. (2019) |
| Studying the Dark Triad of Personality through Twitter Behavior | psychopathy | Preoţiuc-Pietro et al. (2019) |
| Studying the Dark Triad of Personality through Twitter Behavior | Machiavellianism | Preoţiuc-Pietro et al. (2019) |
| Utterance-level Dialogue Understanding: An Empirical Study | Emotion | Ghosal et al. (2020) |

Table 11: Table of all the datasets considered when curating the SOCKET Benchmark.