# OpenReview forum: "Do LLMs Understand Social Knowledge? Evaluating the Sociability of Large Language Models with SocKET Benchmark"
_EMNLP/2023/Conference — EMNLP 2023 Main_

### Official Review · Reviewer_ytnT · 2023-07-25

**Soundness:** 3

**Excitement:**

3: Ambivalent: It has merits (e.g., it reports state-of-the-art results, the idea is nice), but there are key weaknesses (e.g., it describes incremental work), and it can significantly benefit from another round of revision. However, I won't object to accepting it if my co-reviewers champion it.

**Paper Topic And Main Contributions:**

This paper introduces **SOCKET**, a new benchmark with 58 diverse NLP tasks to comprehensively evaluate **social knowledge** in LLMs.

- Tasks are grouped into 5 categories grounded in linguistic theories:
  - Humor & Sarcasm
  - Offensiveness
  - Sentiment & Emotion
  - Social Factors
  - Trustworthiness

The paper benchmarks popular LLMs, finding **limited social capabilities** (~0.7 out of 1 score). It analyzes **correlations** between tasks/categories, showing promise for social knowledge transfer.

## Key Contributions

- Introduces a **systematic benchmark** for social knowledge evaluation
- Provides **empirical analysis** revealing LLMs' weaknesses on social tasks
- Characterizes **relationships** between social language dimensions though task transfer

**Questions For The Authors:**

1. As SOCKET is a theory-grounded collection of datasets, what are the theories behind it?

**Reasons To Accept:**

## Reasons to Accept

1. Comprehensive experiments with a large number of datasets collected on four types of tasks (classification, regression, pairwise, span)

2. Logical writing and clear presentation (well-formatted Tables and Figures)

3. In-depth analysis of experiment results

4. Studies social knowledge transfer which may draw inspiration for the community

**Reasons To Reject:**

1. The selection of tasks is limited to only 4 types which may not be enough.

2. The technical contribution of studying task transfer may be limited, as it is explored in other NLP tasks in prior works

3. More recent state-of-the-art models like GPT-3.5 and GPT-4 were not evaluated.

**Reproducibility:**

4: Could mostly reproduce the results, but there may be some variation because of sample variance or minor variations in their interpretation of the protocol or method.

**Reviewer Confidence:**

3: Pretty sure, but there's a chance I missed something. Although I have a good feel for this area in general, I did not carefully check the paper's details, e.g., the math, experimental design, or novelty.

---

> ### Author Rebuttal · Authors · 2023-08-28
>
> We thank the reviewer for the useful feedback they have provided. We hope to further address and clarify any ambiguity.
>
> 1. We understand the reviewer’s opinion on including more types of NLP tasks in SoCKET for increased generalizability. At the same time, we also would like to emphasize that based on our review, the four types we considered for our benchmark (classification, regression, pairwise comparison, and span detection) were what we found to be most common for NLP tasks related to social aspects. While adding other types of tasks - e.g. summarization, question answering, document ranking - may further enrich our dataset, we believe that the current status of the benchmark is sufficient to test our question of interest for this project: to examine the capability of LLMs for solving NLP tasks requiring social knowledge. We also highlight that most benchmarks contain only a single task type and our aim with SoCKET is to examine the models’ abilities in a wider range of settings than previous work.
>
> 2. As the reviewer righteously points out, the effectiveness of knowledge transfer in LLMs across tasks has been previously studied. However, our study actually shows the opposite result from past work: transfer between tasks requiring social knowledge is **not** as strong as expected and often is non-existent. This result is known and surprising, as social theory suggests we should see a transfer between tasks, e.g., between offensiveness and humor. Our new results highlight the current limitation of LLMs in understanding and effectively learning how to solve tasks requiring social knowledge, signaling a need for stronger models that can encode the social aspects required for solving such tasks. We believe this distinction from prior research makes it a valuable contribution.
>
> 3. Regarding the reviewer’s comment on the addition of recent GPT-based models, we would first like to address that our evaluations on the benchmark were originally intended to only include open-source LLMs to maximize reproducibility. However, given the strong reasoning capabilities of GPT-3.5/GPT-4 and its impact on modern NLP research, we have added an experiment of evaluating SOCKET using gpt-3.5-turbo from OpenAI's API. Interestingly, we were able to discover that while GPT-3.5 outperforms several baseline LLMs, its performance is still only comparable to Google's flan-t5-xxl. This finding strengthens our claim that zero-shot predictions using LLMs largely underperform fine-tuned models. Please bear in mind that while the estimated cost for running GPT-3.5 on SOCKET was within a reasonable price range, the estimated cost for running GPT-4 on the same benchmark was over $700, and unfortunately out of our budget for the period of the rebuttal.
>
> Questions For The Authors:
>
> A summary of the theoretical framework for SOCKET can be found in lines 95-112 of the submitted manuscript. While we were not able to provide a more detailed theoretical grounding due to space constraints, our framework was constructed by identifying several NLP tasks that measure social information and grouping them into relevant categories, where each category is known to play a role in shaping social relationships and interactions in language from perspectives of interpersonal communication (Berger, 2005), psychology and cognitive science (Barret and Lindquist, 2007), and critical studies (Augustinos and Every 2007). We plan to provide further details of the framework construction in an updated version. Please also refer to our response to Reviewer egYJ where we have discussed the motivation behind the framing of "social knowledge".
>
> References
>
> Augoustinos, Martha. and Every, Danielle., 2007. The language of “race” and prejudice: A discourse of denial, reason, and liberal-practical politics. Journal of language and social psychology, 26(2), pp.123-141.
>
> Barrett, Lisa F., Lindquist, Kristin A. and Gendron, Maria., 2007. Language as context for the perception of emotion. Trends in cognitive sciences, 11(8), pp.327-332.
>
> Berger, Charles R., 2005. Interpersonal communication: Theoretical perspectives, future prospects. Journal of communication.

---

### Official Review · Reviewer_CEQi · 2023-08-02

**Soundness:** 4

**Excitement:**

4: Strong: This paper deepens the understanding of some phenomenon or lowers the barriers to an existing research direction.

**Paper Topic And Main Contributions:**

This paper presents a benchmark to evaluate social knowledge covering several theoretically grounded categories. Authors use theoretical grounding to first curate and then combine 58 tasks from 35 previously published datasets. Then, they evaluate the performance of a plethora of popular models in a zero-shot setting as well as finetuning. Experiments show that models perform moderately, at best, and there is significant room for improvement. They also show how task dependencies affect performance and point to possibilities to leverage cross-task transfer.

**Questions For The Authors:**

A: Can you provide more details about finetuning T5 for a regression task? While the prompts are listed in Table 6, other details are incomplete.

B: Unrelated to scores, have you evaluated how GPT3+ models perform on this benchmark? I’m curious to see how LLMs fare (I see you have included BLOOM, OPT etc but it is clear that there is a big gap in performance as compared to GPT models).

C: How can you make sure that cross-task generalization has not been corrupted for large pretrained zero-shot models? What model is used to create Figure 2 (the heatmap)?

D: The results table is unclear and I’m open to revising my review based on clarifications.

**Reasons To Accept:**

Built from 58 tasks across 35 datasets, this benchmark presents a holistic way to analyze models’ understanding of social knowledge. It contains 4 different types of tasks - classification, regression, pairwise comparison and span identification - which somewhat avoids various problems related to task-related artifacts that models often exploit.

The paper points to avenues of leveraging cross-task transfer for improved understanding of social knowledge for models.

**Reasons To Reject:**

What does the AVG in Table 2 mean? What is it an average of? Unifying all the metrics across datasets as an average poses several problems. It washes away performance differences across different columns. How is regression task performance evaluated? How can it be comparable to the other types? No combination of the numbers in Table 2 (for majority) add up to the AVG number.

**Reproducibility:**

3: Could reproduce the results with some difficulty. The settings of parameters are underspecified or subjectively determined; the training/evaluation data are not widely available.

**Reviewer Confidence:**

4: Quite sure. I tried to check the important points carefully. It's unlikely, though conceivable, that I missed something that should affect my ratings.

**Typos Grammar Style And Presentation Improvements:**

The abstract mentions 5 categories but lists only 4. “Social factors” is not listed.
Subsection B.2 does not contain any text.

---

> ### Author Rebuttal · Authors · 2023-08-29
>
> We thank the reviewer for their thoughtful comments and suggestions. We provide our responses to the points addressed to improve the reviewer’s understanding of our work and clarify any ambiguity.
>
> Response to “Reasons to Reject”
>
> The AVG score which we present is an average of the scores for each individual task, which can be found in Table 7.
> The rationale behind using a unified average score is to provide a high-level comparison of the performances of zero-shot and fine-tuned models under various settings, both including task type (regression/classification/pair/span) as well as dimension of social knowledge. Also, it is worth noting that because each task setting has a different number of tasks, it is expected that the numbers in Table 2 when averaged by social dimension or by task type will not equal the overall Avg score. We will clarify the score aggregation in our revised version.
>
> Response to “Questions For The Authors”
>
> The regression tasks are evaluated on the correlation between the predicted and true scores using Pearson’s r. Since Pearson’s r ranges between -1 and 1, we further normalize our score to fit into a range of 0-1 to be comparable with other metrics.
> Regarding the finetuning of T5, we follow the method of Gao et al. (2020), which are mentioned in lines 361-364. We will add more details in future revisions.
>
> While the original focus of our work was to compare the performance of open-source LLMs to advocate open science, we agree with the authors that a comparison of recent GPT models could provide additional insights into understanding the capabilities of modern LLMs. Therefore, we added an experiment using gpt-3.5-turbo on all of the tasks using the same prompts. Our results reveal that GPT-3.5, despite its large number of parameters, is on par with flan-T5-xxl, showing that performance does not always scale with parameter size. We note that we were not able to run GPT-4 on our benchmark during the rebuttal period due to both time and cost constraints (~$700).
>
> There are currently no fully accurate ways to know whether a closed source model (or one where the training data is not known) has seen some examples during its pretraining. However, we recognize that this is a risk also shared with several prior dataset papers testing LLMs. Nevertheless, we highlight that we did not observe any great performance increase for cross-generalization on any of our zero-shot LLM settings. This result suggests that existing models have limited ability in predicting social knowledge in a zero-shot setting, which suggests that even if there had been a leakage of the training data, it is not represented in our results.
>
> Figure 2 was created by obtaining pairwise scores for each setting, which we describe in Section B.6 in the Appendix.
>
> Response to “Typos Grammar Style and Presentation Improvements”
>
> We thank the reviewer for their keen eye in spotting these errors. We have immediately addressed them.

---

### Official Review · Reviewer_egYJ · 2023-08-03

**Soundness:** 3

**Excitement:**

4: Strong: This paper deepens the understanding of some phenomenon or lowers the barriers to an existing research direction.

**Missing References:**

L362: cite ACL version https://aclanthology.org/2021.acl-long.295/

**Paper Topic And Main Contributions:**

**TLDR**: the paper identifies a lack of systematic evaluations targeting social language, proposes a benchmark for detecting social cues to address this gap, evaluates language models on this benchmark, and provides several interesting complementary analyses.

**C1: formulating the problem of social knowledge**
The paper presents an excellent literature review and discussion around challenges of social factors/information/knowledge in NLP systems, motivating this work.

**C2: curating SocKET, a benchmark for evaluating social knowledge**
The paper proposes a benchmark, SocKET, covering 5 categories of social information: humor and sarcasm, offensiveness, sentiment and emotion, trustworthiness, and other social factors (empathy, politeness, intimacy and complaints).
Socket consolidates datasets from 35 existing datasets into 58 evaluations (formulated as classification, regression, pairwise comparison, or span identification), divided along the 5 categories of social information.
An important contribution of this benchmark is its thematic organization of tasks. which enables more comprehensive measures of progress.

**C3: evaluating models and identifying gaps in performance on SocKET**
The paper evaluates several models on SocKET, including zeroshot LLMs on the order of 1-10B parameters (e.g. GPT-J, Alpaca, Bloom, OPT); and finetuned LMs on the order of 10-100M parameters (e.g. bert, roberta). They find that zeroshot models, despite their large number of parameters, perform poorly in contrast to smaller finetuned models. Even finetuned models demonstrate significant room for improvement on this benchmark. Notably, the category of trustworthiness and task of span identification are the most challenging for models.

**C4: presenting interesting findings from complementary analyses**
- zeroshot model size does not typically correlate with performance;
- zeroshot models often do not follow instructions (failing a task by giving an invalid answer rather than a wrong answer);
- finetuned models exhibit notable task dependencies within and across categories.
- finetuned models generally perform worse with multi-task training

**Questions For The Authors:**

Major questions:
A: how are metrics aggregated for categories? If averaged over [tasks/examples], do [smaller/larger] tasks have an outsized effect on performance or does the metric aggregation take into account the number of examples per task?
B: how is class imbalance handled? For instance, the [implicit-hate dataset](https://github.com/SALT-NLP/implicit-hate) is heptuplicated from a single 7-class dataset to seven binary classification datasets. As each class has 10-20% coverage of the dataset, this would imply that the implicit-hate datasets all have 80-90% "no" labels. However, the majority vote results in Table 7 show ~50% performance across these datasets instead of the 80-90% I would have expected from the random split described in A.1?
C: Besides presumably handling class imbalance, what are the preprocessing measures alluded to in A.1? Is e.g. dataset size also normalized or are the sizes reported in Table 1 representative of the final benchmark?

Minor clarifications:
D:  what was meant to be in Appendix B2?
E: [L098] what is meant by "social knowledge is communicated via language understanding"?
F: [L103/110] definition of "appropriacy"?

**Reasons To Accept:**

**TLDR** the paper's value to the community lies in (1) an excellent literature review and discussion that is beneficial to the community; (2) a benchmark which may have the potential to drive progress on important model capabilities; and (3) a series of empirical analyses and findings which are scientifically interesting and pave the way for future work.

I weakly believe the paper should not be accepted in its current state, but nevertheless I believe the work is excellent and very close to being publishable. Primarily, I see the following three reasons to accept:

**A1 Literature review and discussion do a great job of contextualizing an important and under-addressed problem**
The problem of social factors/information/knowledge in language and modern NLP systems is a fascinating topic at the intersection of psychology, linguistics and machine learning. The paper does an excellent job of bringing together various threads of research at this intersection in a cohesive discussion that sheds light on an important open problem in our field.

**A2 Benchmark *potentially* provides a valuable and novel measure of progress on important model capabilities**
While I think our field's relationship with benchmarks is prone to Goodhart's law, I see the value in being able to empirically measure progress on tasks related to capabilities-of-interest; and on a more meta-level, to encourage research in under-explored but important directions. In this regard, the SocKET benchmark provides a compelling reason to accept this paper, insofar as progress on this benchmark is demonstrably tied to progress on the capabilities we care about.

**A3 Empirical analyses raise interesting questions for future research**
In addition to the primary contribution of proposing the SocKET benchmark and evaluating various models on it, the paper also presents several complementary analyses with rigorous experiments and surprising empirical findings (C4). I believe these findings raise open questions that have the potential to spark future research.

**Reasons To Reject:**

**TLDR** the paper has several limitations which may undermine the value of the proposed benchmark to the community, potentially even making it detrimental to progress.

While the paper and work is excellent, I cannot recommend it for acceptance in its current state. I believe benchmarks can be useful if done right, and harmful if done wrong. As of its current state, the paper has no compelling evidence or arguments that progress on this benchmark corresponds to progress on the capabilities used to motivate it. The paper would greatly benefit by incorporating takeaways from the following works:
- [What Will it Take to Fix Benchmarking in Natural Language Understanding?](https://aclanthology.org/2021.naacl-main.385) (Bowman & Dahl, NAACL 2021)
- [Targeting the Benchmark: On Methodology in Current Natural Language Processing Research](https://aclanthology.org/2021.acl-short.85) (Schlangen, ACL-IJCNLP 2021)
- [It Takes Two to Tango: Navigating Conceptualizations of NLP Tasks and Measurements of Performance](https://aclanthology.org/2023.findings-acl.202) (Subramonian et al., Findings 2023)

More specifically, I would recommend this paper for acceptance if the following issues are resolved or adequately addressed:


**R1 Misleading framing of the benchmark**
For me personally, there is a significant disconnect between what is actually measured by the benchmark and how it is framed/motivated in the paper. In particular, I found an important theme in the discussion/references around social information was  language grounded in social interactions (and the associated importance of extra-linguistic context and pragmatic inference). While each category in the socket benchmark is justified and supported by references as being a facet of social information, the relevance of these tasks to the original question of "social knowledge" seems tenuous. I'm not convinced the benchmark is theory-driven/grounded as much as it is superficially and post-hoc linked to various theories.

In the same way that a new sentiment analysis benchmark would not really help our understanding of "social knowledge" (even though sentiment is easily framed as an important facet of social information), I don't see how benchmarks for predicting humor/offensiveness/trustworthiness/etc contribute to the problems motivating this paper.
I still believe the benchmark is valuable in-and-of-itself, but I feel there was a bait and switch between Sections 1-2 where the problem of "social knowledge" is framed, and Section 3 where the benchmark is introduced.

I believe the framing briefly presented in the conclusion (i.e. "social cue detection") is much more representative of the benchmark. A discussion situating (the narrower focus of) social cue detection within (the broader scope of) social knowledge/information/factors is missing and would improve the paper I believe. I could see a mischaracterization of this benchmark as "evaluating sociability" or "understanding social knowledge" may lead to misleading impressions of progress, especially given the chasm between the task formulations (classification, regression, etc on non-conversational examples) and how LLMs are used in practice (natural dialogue). I think the value of the benchmark is compelling without needing to over-hype it.

**R2 Aggregating of results and limited interpretability**
While I see the value of combining thematically related tasks into aggregate evaluations, I believe there are potential pitfalls to only reporting a single metric per category. A finer grained understanding of where models succeed, fail, or have high variance would be invaluable in contextualizing progress on this benchmark. Table 7 is a good start, but it is only briefly referenced in the context of multi-task training without any discussion of the stratified results. Without this kind of finer grained analysis, I'm worried progress on this benchmark could end up being quite misleading. I'm particularly worried about this, because the benchmark brings together various existing datasets which may have important systematic differences or limitations which skew or bias evaluation results on this benchmark. More generally, this kind of aggregating makes it difficult to interpret what an improvement in any given category actually means. For instance, as a toy example, improving "humour and sarcasm" may be improving "humor" or "sarcasm"; but this is lost in the aggregate metric.

**Reproducibility:**

2: Would be hard pressed to reproduce the results. The contribution depends on data that are simply not available outside the author's institution or consortium; not enough details are provided.

**Reviewer Confidence:**

4: Quite sure. I tried to check the important points carefully. It's unlikely, though conceivable, that I missed something that should affect my ratings.

**Typos Grammar Style And Presentation Improvements:**

L428: Figure 1 shows

Table 6: there is a mismatch between the prompt for certain REG tasks and the cited method of  [Gao et al. 2021](https://aclanthology.org/2021.acl-long.295/) (i.e. "The score should be ranging from 0.0 to 5.0, and can be a decimal" versus measuring the probability of "yes" vs "no" as a proxy for regression).

---

> ### Author Rebuttal · Authors · 2023-08-29
>
> The authors greatly appreciate the attention paid to the work and the clear amount of thought put into this rigorous and nuanced review. We’re excited to be given the opportunity to respond to some of the excellent feedback we have received in this review. Your review excited us, so please excuse the length as we want to deeply engage with your comments and hope our response continues the great conversation.
>
> Framing:
>
> We fully agree that accurate framing is crucial to the development of work that attempts to explore new grounds for the field, and that the downstream consequences of improper framing are numerous.
>
> The choice of the term “social knowledge” as framing came from a desire to not be overly prescriptive about the specific nature of signals being recognized by models. The benchmark aims to estimate the capacity of LLMs to pick up on a variety of constructs that literature in Psychology and Linguistics have shown to be central to the social-functional use of language, including norm-adherence, power relations, sentiment analysis, and more. We ultimately selected the term “social knowledge” because it has been used previously as a broad category in psychology (e.g., Turiel 1983; Adolphs 2009) and matched the capabilities we are interested in. We will provide more details and motivation for this broad notational framing in revisions.
>
> We agree on the need to improve the framed representation for the benchmark. Respectfully, we feel that the term “social cues” alone might be too limiting of a framework due to a perceived emphasis on in-domain syntax. However, we do agree that the framing of the benchmark of the text should be adjusted—especially following insights gained from the Bowman and Dahl article supplied by the reviewer. We are happy to take suggestions for revisions that can more accurately represent the capabilities and limitations of the benchmark.
>
> The reviewer notes a gap between the Theory described in Section 2 and Tasks motivated in Section 3. We appreciate this feedback and will work to close the gap. Our definition of “social knowledge” is broad and we highlight key aspects of social communication pointed to in research from Communications, Psychology, and Linguistics (Lines 95-112; throughout Section 3). Of course, we fully admit these dimensions are not the totality of all social knowledge (lines 709-814), but they are a start. We also fully agree with the reviewer’s notion that “a new sentiment analysis benchmark would not really help our understanding of ‘social knowledge’” but point out that the goal is not proposing five _separate_ categories but five _interrelated_ categories (Lines 191-214). Our categories have predicted interactions from theory and by including all in the same benchmark, we can test whether models benefit from these implicit relationships—in fact, we don’t see this (Section 5), pointing to a clear gap in current performance!
>
> Aggregation:
>
> We agree with the reviewer’s perceived tension around data aggregation. Too much aggregation obscures real progress and differences between models, while too little aggregation can lead to too many scores and uncertainty around which scores need to go up, making comparison difficult—or worse, excessive benchmark hill-climbing around many disaggregated scores.
> Our choice to aggregate was based on current practice from previous NLP benchmarks such as Glue (Wang et al., EMNLP 2018), SuperGLUE (Wang et al., NeurIPS 2019), and MTEB (Muennighoff et al., EACL 2023). All of these benchmarks aggregate dissimilar tasks into a single metric that is a proxy for performance at some larger construct like “Language Understanding”. For example, in the MTEB paper the authors average the performances of tasks such as classification, clustering, semantic textual similarity (STS), and summarization into a single metric despite their dissimilarity. While such aggregating does minimize disparities between these tasks when only the single aggregate metric is reported, the single metric does provide utility for comparing models.
> SoCKET presents a challenging case for aggregation as it has more than 50 different tasks. We decided it would be inconvenient to report the performance of every combination in the main paper. Also, emphasizing the comparisons between different models at an individual task level can wrongly introduce the notion of promoting “leaderboard chasing” where teams can highlight how their models achieve #1 for a particular set of tasks. Our aggregation of tasks for each social category is intended to instead focus on the aspect of measuring a model’s general social knowledge capacity.
>
> However, we fully agree with the spirit of the reviewer’s comment. In Table 2, we report category-level aggregated information, e.g., for “Trustworthy” tasks, and for tasks for each type, e.g., pair comparison. We intend that any users of the benchmark would report these slightly disaggregated metrics as well to show which types of social knowledge are recognized. We will certainly add more details around the experiments from Table 7 to highlight the importance of per-task learning.
> We also agree that the addition of finer-grained analyses on where models vary in performance and behave expectedly/unexpectedly can improve the quality of our paper. For instance, one experiment that we did not include in the main paper was that LLMs have varying capabilities in understanding the instructions of the tasks (Figure 3). When reporting the performance of LLMs on tasks, it could be beneficial to differentiate the performance of (1) where the model failed to understand the task and (2) where the model made an incorrect prediction. While such additional experiments are out of the scope of this paper, we believe subsequent research using this benchmark can be very useful.
>
> Questions For The Authors:
>
> Major Questions
> A: Metrics are aggregated for categories as averages of F1 scores. Each task is scored independently with a macro F1, and the unweighted mean of those scores makes up the aggregate score for a category.
>
> B: In an attempt to stay consistent and ensure fair comparison across several different task types—some of which, like the implicit-hate-dataset, emphasize realistic ratios of positively-labeled hate speech and negative labels—we use a unified metric (macro F1) to measure performance on each task. Thus, for example, in the implicit-hate-dataset, stereotypical_hate makes up 17.9% of all implicit hate samples, which account for 28% of the dataset. This would suggest that ~5% of the implicit-hate-dataset samples in our test set should be positive for stereotypical_hate, and this is what we find. Calculating the macro F1 for a set of ground truth labels with 5% positive and the rest negative and the estimated labels being all negative we find that the score is ~0.49, as reported.
>
> Minor Questions:
> D: B.2 was meant to introduce Table 4 in the appendix and it has been amended. Thank you for bringing that to our attention.
>
> E: By this we mean that information regarding the social-functional aspects of interpersonal communication, which we consider to be under the umbrella term “social knowledge”, are transmitted (perhaps not exclusively) via language use.. This includes those studies, for example, in cognitive science regarding the role of language in emotion perception (Barrett et al., 2007), in interpersonal communication regarding predictability in language being used to assert relationship status in social interaction (Berger, 2005), and in social psychology showing language use conveying implicit racial prejudices of the speaker (Augoustinos and Every, 2007) among others.
>
> F: “Appropriacy” in this context comes specifically from Eggins, 2004 (below) and can be found in other texts (see Allen, 2007) as being the forces which cause language-users to choose certain words or phrases instead of all the possible alternatives.
>
>
>
>
> Missing References:
> The ACL version of “Making Pre-trained Language Models Better Few-shot Learners” by Gao, Fisch, and Chen has been cited, replacing the preprint previously cited.
>
> Typos Grammar Style And Presentation Improvements:
> We are sorry for the confusion. Gao’s method was specifically used for fine-tuning the t5 model for regression tasks. The listed prompts without “yes” and “no” was used across all the other zeroshot prompting experiments. We will clarify this in future revisions.
>
> Reproducibility:
> In regard to the reproducibility score, the authors would like to emphasize that all datasets and models used in the paper are open-sourced. Moreover, the benchmark dataset will be publicized when ACL anonymity rules allow, and preprocessing scripts will be made available to reconstruct the benchmark from the original datasets collected. If there are any additional barriers to replication, we would greatly appreciate the suggestions, as we recognize that consistency and reproducibility are at the heart of any useful benchmark.
>
> References
>
> Adolphs, Ralph., 2009. The social brain: neural basis of social knowledge. Annual review of psychology, 60, pp.693-716.
>
> Allan, Keith., 2007. The pragmatics of connotation. Journal of pragmatics, 39(6), pp.1047-1057.
>
> Augoustinos, Martha. and Every, Danielle., 2007. The language of “race” and prejudice: A discourse of denial, reason, and liberal-practical politics. Journal of language and social psychology, 26(2), pp.123-141.
>
> Barrett, Lisa F., Lindquist, Kristin A. and Gendron, Maria., 2007. Language as context for the perception of emotion. Trends in cognitive sciences, 11(8), pp.327-332.
>
> Berger, Charles R., 2005. Interpersonal communication: Theoretical perspectives, future prospects. Journal of communication.
>
> Eggins, Suzanne., 2004. Introduction to systemic functional linguistics. A&c Black.
>
> Muennighoff et al., 2023. MTEB: Massive Text Embedding Benchmark. EACL
>
> Turiel, Elliot., 1983. The development of social knowledge: Morality and convention. Cambridge University Press.
>
> Wang et al., 2018. GLUE: A Multi-Task Benchmark and Analysis Platform for Natural Language Understanding. EMNLP
>
> Wang et al. 2019. Superglue: A stickier benchmark for general-purpose language understanding systems. NeurIPS

---

### Meta-Review · Area_Chair_vquG · 2023-09-10

**Recommendation:** 5

**Metareview:**

This paper presents a new theory-driven benchmark, SOCKET, that contains 58 NLP tasks testing social knowledge, grouped into five categories: humor & sarcasm, offensiveness, sentiment & emotion, and trustworthiness.
LLMs are evaluated on this benchmark, providing interesting findings.

While not all the more recent LLMS have been evaluated, the benchmark is the focus of the paper and the findings are already interesting.
While very interesting, the sudy of the social capability transfer could be improve, maybe in a future work.

---

### Decision · Program_Chairs · 2023-10-07

**Decision:**

Accept-Main

**Comment:**

This paper presents a new theory-driven benchmark, SOCKET, that contains 58 NLP tasks testing social knowledge, grouped into five categories: humor & sarcasm, offensiveness, sentiment & emotion, and trustworthiness.
LLMs are evaluated on this benchmark, providing interesting findings.

While not all the more recent LLMS have been evaluated, the benchmark is the focus of the paper and the findings are already interesting.
While very interesting, the sudy of the social capability transfer could be improve, maybe in a future work.